# STING mediates immune responses in the closest living relatives of animals

Arielle Woznica[1]*, Ashwani Kumar[2], Carolyn R Sturge[1], Chao Xing[2], Nicole King[3], Julie K Pfeiffer[1]*

[1]Department of Microbiology, University of Texas Southwestern Medical Center, Dallas, United States; [2]McDermott Center Bioinformatics Lab, University of Texas Southwestern Medical Center, Dallas, United States; [3]Howard Hughes Medical Institute, and Department of Molecular and Cell Biology, University of California, Berkeley, Berkeley, United States

**Abstract** Animals have evolved unique repertoires of innate immune genes and pathways that provide their first line of defense against pathogens. To reconstruct the ancestry of animal innate immunity, we have developed the choanoflagellate *Monosiga brevicollis*, one of the closest living relatives of animals, as a model for studying mechanisms underlying pathogen recognition and immune response. We found that *M. brevicollis* is killed by exposure to *Pseudomonas aeruginosa* bacteria. Moreover, *M. brevicollis* expresses STING, which, in animals, activates innate immune pathways in response to cyclic dinucleotides during pathogen sensing. *M. brevicollis* STING increases the susceptibility of *M. brevicollis* to *P. aeruginosa*-induced cell death and is required for responding to the cyclic dinucleotide 2'3' cGAMP. Furthermore, similar to animals, autophagic signaling in *M. brevicollis* is induced by 2'3' cGAMP in a STING-dependent manner. This study provides evidence for a pre-animal role for STING in antibacterial immunity and establishes *M. brevicollis* as a model system for the study of immune responses.

*For correspondence:
Arielle.Woznica@
UTSouthwestern.edu (AW);
Julie.Pfeiffer@UTSouthwestern.
edu (JKP)

Competing interest: The authors declare that no competing interests exist.

## Editor's evaluation

This is a very exciting story on the ancient role that STING plays in microbial infections.

## Introduction

Innate immunity is the first line of defense against pathogens for all animals, in which it is crucial for distinguishing between self and non-self, recognizing and responding to pathogens, and repairing cellular damage. Some mechanisms of animal immunity have likely been present since the last common eukaryotic ancestor, including RNAi, production of antimicrobial peptides, and the production of nitric oxide (*Shabalina and Koonin, 2008*; *Richter and Levin, 2019*). However, many gene families that play critical roles in animal innate immune responses are unique to animals (*Richter et al., 2018*).

Comparing animals with their closest relatives, the choanoflagellates, can provide unique insights into the ancestry of animal immunity and reveal other key features of the first animal, the 'Urmetazoan' (*King et al., 2008*; *Brunet and King, 2017*; *Richter and King, 2013*). Choanoflagellates are microbial eukaryotes that live in diverse aquatic environments and survive by capturing and phagocytosing environmental bacteria (*Leadbeater, 2015*) using their 'collar complex', an apical flagellum surrounded by actin-filled microvilli (*Figure 1A*; *Leadbeater, 2015*; *Dayel and King, 2014*). Several innate immune pathway genes once considered to be animal-specific are present in choanoflagellates, including cGAS and STING, both of which are crucial for innate responses to cytosolic DNA in animals (*Figure 1—figure supplement 1*; *Richter et al., 2018*; *Wu et al., 2014*; *Levin et al., 2014*).

Although the phylogenetic distribution of these gene families reveals that they first evolved before animal origins, their functions in choanoflagellates and their contributions to the early evolution of animal innate immunity are unknown.

STING (stimulator of interferon genes) is a signaling protein that activates innate immune responses to cytosolic DNA during bacterial or viral infection (*Ablasser and Chen, 2019*; *Ablasser et al., 2013*; *Ahn and Barber, 2019*). Although STING homologs are conserved in diverse invertebrate and vertebrate animals (reviewed in *Margolis et al., 2017*; *Wu et al., 2014*; *Margolis et al., 2017*; *Kranzusch et al., 2015*), mechanisms of STING activation are best understood in mammals. In mammals, STING is activated by binding 2'3' cGAMP, an endogenous cyclic dinucleotide produced by the sensor cGAS (cyclic GMP-AMP synthase) upon detecting cytosolic DNA (*Ishikawa and Barber, 2008*; *Sun et al., 2013*; *Burdette et al., 2011*; *Diner et al., 2013*; *Gao et al., 2013*; *Ablasser et al., 2013*). In addition, cyclic dinucleotides produced by bacteria can also activate STING (*Burdette et al., 2011*; *Moretti et al., 2017*). Importantly, STING domain-containing systems are present in bacteria where they may contribute to anti-phage defense (*Cohen et al., 2019*; *Morehouse et al., 2020*), raising the possibility that eukaryotic STING-like proteins were acquired from lateral gene transfer (*Burroughs et al., 2020*). Comparative genomics suggests that STING domains arose at least three independent times in eukaryotes, including once in the stem lineage leading to Choanozoa, the clade containing animals and choanoflagellates (*Burroughs et al., 2020*).

Choanoflagellates have already served as powerful models for studying the origin of animal multicellularity and cell differentiation (*Levin et al., 2014*; *Alegado et al., 2012*; *Brunet et al., 2021*; *Dayel et al., 2011*; *Woznica et al., 2017*; *Brunet et al., 2019*; *Laundon et al., 2019*) and are ideally positioned to yield insights into the evolution of animal immune pathways. Therefore, we sought to establish the choanoflagellate *Monosiga brevicollis* as a model for studying pathogen recognition and immune responses. Here, we report that *Pseudomonas aeruginosa* bacteria are pathogenic for *M. brevicollis*. Through our study of interactions between *P. aeruginosa* and *M. brevicollis*, we determined that STING functions in the choanoflagellate antibacterial response. In addition, we demonstrate that STING is necessary for mediating responses to the STING agonist 2'3' cGAMP in vivo, and that 2'3' cGAMP induces STING-dependent autophagic signaling. Our results demonstrate that key features of STING-mediated immune responses are conserved in *M. brevicollis*, thereby expanding our understanding of the pre-metazoan ancestry of STING signaling.

## Results

### *P. aeruginosa* has pathogenic effects on *M. brevicollis*

One impediment to studying immune responses in choanoflagellates has been the lack of known choanoflagellate pathogens. While bacteria are obligate prey and can regulate mating, multicellular development, and cell contractility in choanoflagellates, to our knowledge no bacteria with pathogenic effects have been described (*Alegado et al., 2012*; *Woznica et al., 2017*; *Brunet et al., 2019*; *Woznica et al., 2016*; *Ireland et al., 2020*; *Hake, 2021*). For this study, we focused on the choanoflagellate *Monosiga brevicollis,* which has a sequenced genome (*King et al., 2008*) and grows robustly under laboratory conditions in co-culture with *Flavobacterium* prey bacteria (*Brunet and King, 2017*). To identify potential pathogens of choanoflagellates, we screened select bacteria – including environmental isolates and known animal pathogens and commensals (*Table 1*) – to test whether any of these induced *M. brevicollis* behavioral changes or reduced cell survival.

After co-culturing *M. brevicollis* with bacteria for 24 hr, only the gammaproteobacterium *Pseudomonas aeruginosa,* a ubiquitous environmental bacterium and opportunistic pathogen of diverse eukaryotes (*Moradali et al., 2017*; *Mahajan-Miklos et al., 1999*; *Pukatzki et al., 2002*; *Rahme et al., 1997*), altered the behavior and growth dynamics of *M. brevicollis*. Under standard laboratory conditions, *M. brevicollis* is a highly motile flagellate and swims up in the water column (*Video 1*, *Supplementary file 1*). However, after 12–14 hr in the presence of *P. aeruginosa* strains PAO1 and PA14, a large proportion of *M. brevicollis* cells settled to the bottom of the culture dish (*Video 1*, *Supplementary file 1*). Immunofluorescence staining revealed that *M. brevicollis* cells exposed to *P. aeruginosa* had truncated flagella compared to cells exposed to *E. coli* or other bacteria that did not induce cell settling (*Figure 1B*). To determine the effects of *P. aeruginosa* on cell viability, we added *P. aeruginosa* strain PAO1 or control gammaproteobacteria to *M. brevicollis* and monitored cell density over the

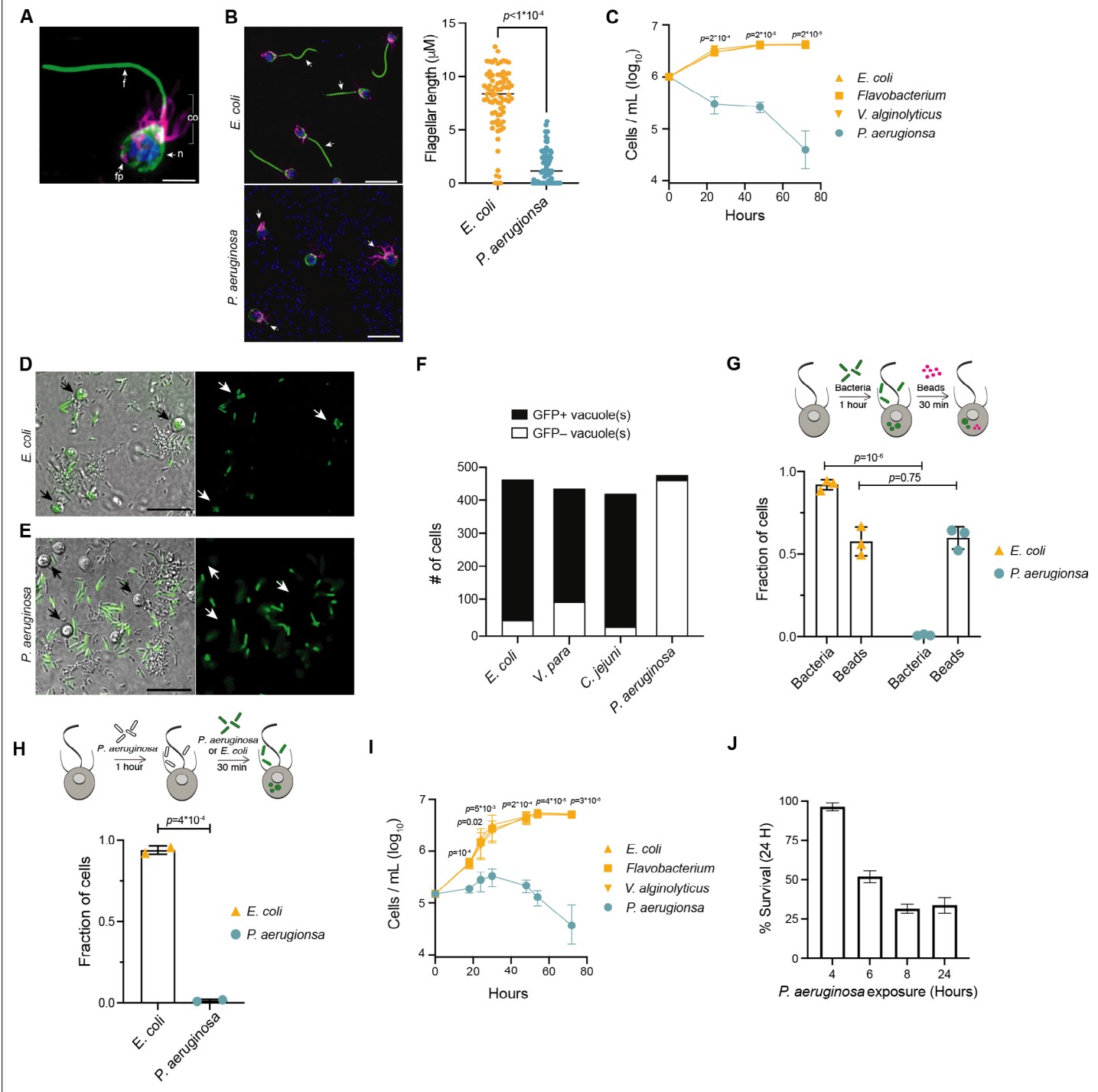

**Figure 1.** *P. aeruginosa* has pathogenic effects on *M. brevicollis*. (**A**) Immunofluorescence illuminates the diagnostic cellular architecture of *M. brevicollis*, including an apical flagellum (**f**) made of microtubules, surrounded by an actin-filled microvilli feeding collar (co). Staining for tubulin (green) also highlights cortical microtubules that run along the periphery of the cell body, and staining for F-actin (magenta) highlights basal filopodia (fp). DNA staining (blue) highlights the nucleus (**n**). (**B**) *M. brevicollis* exhibits truncated flagella after exposure to *P. aeruginosa*. *M. brevicollis* were exposed to *E. coli* or *P. aeruginosa* for 24 hr, and then fixed and immunostained. Arrows point to flagella. Green: anti-tubulin antibody (flagella and cell body), magenta: phalloidin (collar), blue: Hoechst (bacterial and choanoflagellate nuclei). Scale bars represent 10 μm. Flagellar length was quantified using Fiji, and statistical analysis (unpaired t-tests) was performed in GraphPad software. (**C**) Exposure to *P. aeruginosa*, but not other Gammaproteobacteria, results in *M. brevicollis* cell death. Bacteria were added to *M. brevicollis* culture at an MOI of 1.5 (at Hours = 0), and *M. brevicollis* cell density was quantified at indicated time points. Data represent mean ± SD for three biological replicates. Statistical analysis (multiple unpaired t-tests) was performed in GraphPad software; p-values shown are from comparisons between *Flavobacterium* and *P. aeruginosa*. (**D–F**) *M. brevicollis* does not ingest

*Figure 1 continued*

*P. aeruginosa* bacteria. (**D,E**) *M. brevicollis* were fed either fluorescent *E. coli* (**D**) or *P. aeruginosa* (**E**) for 1 hr, and then visualized by DIC (D,E, left) and green fluorescence (D, E, right). Fluorescent food vacuoles were observed in choanoflagellates fed *E. coli*, but not *P. aeruginosa*. (**F**) *M. brevicollis* was exposed to GFP-expressing *E. coli*, *V. parahaemolyticus*, *C. jejuni*, or *P. aeruginosa* (MOI = 50) for 1 hr, and then imaged by DIC and green fluorescence to quantify number of cells with internalized bacteria. Choanoflagellate cells with ≥1 GFP+ food vacuole were scored as GFP+, and cells without any GFP+ food vacuoles were scored as GFP–. Data represent cells quantified over three biological replicates. (**G,H**) *P. aeruginosa* does not broadly inhibit *M. brevicollis* phagocytosis. (**G**) Internalization of 0.2 µm fluorescent beads was used to quantify phagocytic activity after exposure to *E. coli* or *P. aeruginosa* bacteria. Although cells did not phagocytose *P. aeruginosa,* cells exposed to *E. coli* and *P. aeruginosa* had similar phagocytic uptake of beads. Data represent n = 600 cells from three biological replicates. Statistical analyses (multiple unpaired t-tests) were performed in GraphPad software. (**H**) Exposure to *P. aeruginosa* does not inhibit phagocytic uptake of *E. coli*. Internalization of fluorescent *E. coli* or *P. aeruginosa* bacteria was quantified after exposure to unlabeled *P. aeruginosa* (PAO1 strain). Data represent n = 200 cells from two biological replicates. Statistical analysis (unpaired t-test) was performed in GraphPad software. (**I**) Secreted *P. aeruginosa* molecules are sufficient to induce *M. brevicollis* cell death. 5 % (vol/vol) bacterial conditioned medium was added to *M. brevicollis* culture (at Hours = 0), and *M. brevicollis* cell density was quantified at indicated time points. Data represent mean ± SD for three biological replicates. Statistical analysis (multiple unpaired t-tests) was performed in GraphPad software, and *p*-values shown are from comparisons between *Flavobacterium* and *P. aeruginosa*. (**J**) Sustained exposure to secreted *P. aeruginosa* molecules is required to induce *M. brevicollis* cell death. *P. aeruginosa* or *Flavobacterium* conditioned medium (5% vol/vol) was added to stationary-phase *M. brevicollis* cultures. After indicated times, cultures were washed and resuspended in fresh media. *M. brevicollis* cell density was quantified after 24 hr. The % survival is a measure of the cell density of *P. aeruginosa*-treated cells relative to *Flavobacterium*-treated controls. Data represent mean ± SD for three biological replicates.

The online version of this article includes the following figure supplement(s) for figure 1:

**Figure supplement 1.** Presence of animal innate immune genes in choanoflagellates.

course of 72 hr (*Figure 1C*). While *M. brevicollis* continued to proliferate in the presence of control gammaproteobacteria, exposure to *P. aeruginosa* PAO1 resulted in cell death.

Choanoflagellates prey upon bacteria and ingest them through phagocytosis (*Leadbeater, 2015*; *Dayel and King, 2014*). However, many bacterial pathogens, including *P. aeruginosa*, have evolved strategies to prevent or resist phagocytosis by eukaryotic cells (*Uribe-Querol and Rosales, 2017*; *Yoon et al., 2018*). Therefore, we examined whether phagocytosis of *P. aeruginosa* is required to induce cell death. To track phagocytosis, we added GFP-expressing *E. coli* DH5α (*Figure 1D*) or *P. aeruginosa* PAO1 (*Figure 1E*) to *M. brevicollis* and monitored the cultures by live imaging. After one hour, while 92 % of *M. brevicollis* cells incubated with *E. coli* –GFP had GFP+ food vacuoles, only 3 % of cells incubated with PAO1-GFP had GFP+ food vacuoles (*Figure 1F*). *M. brevicollis* also robustly phagocytosed GFP-expressing *V. parahaemolyticus* and *C. jejuni* (*Figure 1F*).

Next, to determine if *P. aeruginosa* broadly disrupts *M. brevicollis* phagocytosis, which could induce cell death through starvation, we incubated *M. brevicollis* with GFP-expressing PAO1 or GFP-expressing *E. coli* for one hour, and then added 0.2 mm fluorescent beads for an additional 30 min as an independent measure of phagocytic activity. The fraction of *M. brevicollis* cells with internalized 0.2 mm beads was similar in cultures incubated with *E. coli* DH5αand PAO1 (*Figure 1G*). Moreover, exposure to *P. aeruginosa* did not inhibit phagocytic uptake of *E. coli* (*Figure 1H*). These results suggest that exposure to *P. aeruginosa* does not broadly inhibit phagocytosis.

The above results suggested that the pathogenic effects of *P. aeruginosa* are induced by factors secreted by extracellular bacteria. In addition, diverse secreted bacterial molecules have been previously shown to influence choanoflagellate cell biology (*Alegado et al., 2012*; *Woznica et al., 2017*; *Woznica et al., 2016*). Therefore, we next investigated the effects of secreted *P. aeruginosa* molecules on *M. brevicollis* viability. Exposure of *M. brevicollis* to conditioned medium from *P. aeruginosa* PAO1 or diverse non-pathogenic gammaproteobacteria revealed that PAO1 conditioned medium is sufficient to restrict growth and induce cell death (*Figure 1I*). Similar to live bacteria, exposure to *P. aeruginosa* conditioned medium led to reduced motility and truncated flagella in *M. brevicollis* after approximately 8–10 hr.

Because numerous *P. aeruginosa* secreted virulence factors have been characterized (*Moradali et al., 2017*; *Klockgether and Tümmler, 2017*), we screened a battery of isogenic PAO1 strains with deletions in known virulence genes to determine if any of these factors contribute to the pathogenic effects on *M. brevicollis* (*Table 2*). All strains tested induced similar levels of *M. brevicollis* cell death as the parental PAO1 strain, suggesting that none of the deleted virulence genes alone are essential for inducing cytotoxicity in *M. brevicollis*. The bioactivity in the conditioned media was also found to be heat, protease, and nuclease resistant, indicating that the virulence factors are unlikely

**Table 1.** Bacteria screened for pathogenic effects.

| Bacterium | Pathogenic effects | Reference or details | Source |
|---|---|---|---|
| *Aeromonas hyrophila* | – | Environmental isolate | This study |
| *Bacillus aquimaris* | – | Environmental isolate | This study |
| *Bacillus badius* | – | Mouse isolate | Julie Pfeiffer |
| *Bacillus cereus* | – | Environmental isolate | This study |
| *Bacillus indicus* | – | Environmental isolate | This study |
| *Bacillus marisflavi* | – | Environmental isolate | This study |
| *Bacillus pumilus* | – | Mouse isolate | Julie Pfeiffer |
| *Bacillus safensis* | – | Mouse isolate | Julie Pfeiffer |
| *Bacillus subtilus* | – | ATCC 6633 | Julie Pfeiffer |
| *Bacteroides acidifaciens* | – | Mouse isolate | Julie Pfeiffer |
| *Burkholderia multivorans* | – | ATCC 17616 | David Greenberg |
| *Campylobacter jejuni* GFP | – | DRH3123 | David Hendrixson |
| *Deinococcus* sp. | – | Environmental isolate | This study |
| *Enterococcus cloacae* | – | Mouse isolate | Julie Pfeiffer |
| *Enterococcus faecium* | – | Mouse isolate | Julie Pfeiffer |
| *Escherichia coli* BW25113 | – | *Datsenko and Wanner, 2000* | David Greenberg |
| *Escherichia coli* DH5a GFP | – | | David Hendrixson |
| *Escherichia coli* ECC-1470 | – | *Leimbach et al., 2015* | Julie Pfeiffer |
| *Escherichia coli* K12 | – | ATCC 10798 | Julie Pfeiffer |
| *Flavobacterium sp.* | – | *King et al., 2008* | Isolated from ATCC PRA-258 |
| *Lactobacillus johnsonii* | – | Mouse isolate | Julie Pfeiffer |
| *Pseudoalteromonas sp.* | – | Environmental isolate | This study |
| *Pseudomonas aeruginosa* PA-14 | + | *Rahme et al., 1995* | Andrew Koh |
| *Pseudomonas aeruginosa* PAO1 | + | ATCC 15692 | David Greenberg |
| *Pseudomonas aeruginosa* PAO1-GFP | + | *Bloemberg et al., 1997* | David Greenberg |
| *Pseudomonas granadensis* | – | Environmental isolate | This study |
| *Staphylococcus aureus* | – | ATCC 23235 | Julie Pfeiffer |
| *Staphylococcus sp.* | – | Mouse isolate | Julie Pfeiffer |
| *Vibrio alginolyticus* | – | Environmental isolate | Kim Orth |
| *Vibrio furnissii* | – | Environmental isolate | This study |
| *Vibrio parahaemolyticus* | – | Environmental isolate | This study |
| *Vibrio parahaemolyticus* RimD-GFP | – | *Ritchie et al., 2012* | Kim Orth |
| *Vibrio ruber* | – | Environmental isolate | This study |
| *Vibrio sp.* | – | Environmental isolate | This study |

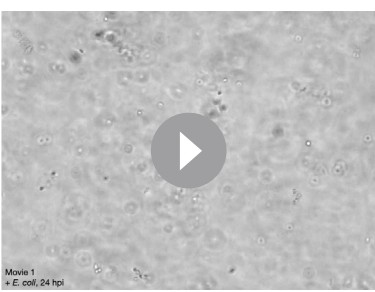

**Video 1.** *Supplementary file 1*. *P. aeruginosa* influences *M. brevicollis* motility. Movies depicting *M. brevicollis* cultures after exposure to *E. coli* or *P. aeruginosa* bacteria for 16 hours. In the absence of pathogenic bacteria,*M. brevicollis* is a highly motile flagellate and swims up in the water column (Movie 1). However, co-culturing *M. brevicollis* with *P. aeruginosa* results in reduced motility and cell settling (Movie 2). https://elifesciences.org/articles/70436/figures#video1

to be proteins or nucleic acids (*Table 3*). In addition, we found that subjecting the conditioned media to methanol extraction followed by liquid chromatography-mass spectrometry resulted in specific fractions that recapitulated the bioactivity of the conditioned media. Although further detailed chemical analysis is required to determine the molecular nature of these factors, these data indicate that secreted *P. aeruginosa* small molecules are sufficient for inducing cell death in *M. brevicollis*.

Finally, we investigated whether transient exposure to *P. aeruginosa* is sufficient to induce cell death in *M. brevicollis* (*Figure 1J*). Because choanoflagellates likely experience fluctuations in the chemical composition of aquatic microenvironments on various timescales, we exposed stationary-phase *M. brevicollis* to *P. aeruginosa* conditioned media for increasing durations, and then assessed survival relative to *Flavobacterium*-treated controls after 24 hr. While *M. brevicollis* cell death was not observed after treatment with *P. aeruginosa* conditioned media for 4 hr or less, exposures lasting for 6 hr or longer reduced *M.*

**Table 2.** *P. aeruginosa* deletion strains.

| | | | Effects on *M. brevicollis* | |
| | | | Truncated Flagellum/ | |
| Strain name | Gene | Putative ORF function | Settling | Cell Death |
|---|---|---|---|---|
| MPAO1 | | parent to library stain | + | + |
| PW5035 | pvdE | pyoverdine biosynthesis protein PvdE | + | + |
| PW5034 | pvdE | pyoverdine biosynthesis protein PvdE | + | + |
| PW1059 | exoT | exoenzyme T | + | + |
| PW3078 | toxA | exotoxin A precursor | + | + |
| PW3079 | toxA | exotoxin A precursor | + | + |
| PW4736 | exoY | adenylate cyclase ExoY | + | + |
| PW4737 | exoY | adenylate cyclase ExoY | + | + |
| PW6886 | rhlA | rhamnosyltransferase chain A | + | + |
| PW6887 | rhlA | rhamnosyltransferase chain A | + | + |
| PW7478 | exoS | exoenzyme S | + | + |
| PW7479 | exoS | exoenzyme S | + | + |
| PW7303 | lasB | elastase LasB | + | + |
| PW7302 | lasB | elastase LasB | + | + |
| PW3252 | aprA | alkaline metalloproteinase precursor | + | + |
| PW3253 | aprA | alkaline metalloproteinase precursor | + | + |
| PW4282 | lasA | LasA protease precursor | + | + |
| PW4283 | lasA | LasA protease precursor | + | + |
| RP436 | popB | T3SS translocase | + | + |
| RP576 | exoS, exoT, exoY | T3SS effector molecules | + | + |

**Table 3.** *M. brevicollis* response to *P. aeruginosa* factors.

| Treatment | Cell Death | Interpretation |
| --- | --- | --- |
| Live *P. aeruginosa* | ++ | |
| *P. aeruginosa* conditioned media (CM) | ++ | Factor(s) are secreted by *P. aeruginosa* |
| *P. aeruginosa* outer membrane vesicles | – | Factor(s) are not present in outer membrane vesicles |
| CM, boiled 20 min | ++ | Factor(s) are not heat labile |
| CM+ proteinase K, followed by 80 C for 30 min | ++ | Factor(s) are not proteins |
| CM+ DNAse and RNAse | ++ | Factor(s) are not nucleic acids |
| CM MeOH extraction | ++ | Factor(s) are organic compounds |

*brevicollis* survival (*Figure 1J*). These results suggest that cell death pathways are not induced immediately in response *P. aeruginosa* virulence factors, but are activated after longer exposures to *P. aeruginosa* conditioned media.

## Upregulation of *M. brevicollis* STING in response to *P. aeruginosa*

To identify potential genetic pathways activated by *M. brevicollis* in response to pathogenic bacteria, we performed RNA-seq on *M. brevicollis* exposed to conditioned medium from either *P. aeruginosa* or *Flavobacterium sp.*, the non-pathogenic bacterial strain used as a food source (*Table 1*). We found that 674 genes were up-regulated and 232 genes were downregulated twofold or greater (FDR ≤ 10$^{-4}$) upon *P. aeruginosa* exposure compared to cells exposed to *Flavobacterium* (*Figure 2A*). The upregulated genes were enriched in biological processes including response to stress, endocytosis, microtubule-based movement, mitochondrial fission, and carbohydrate metabolism. Genes down-regulated in response to *P. aeruginosa* were enriched in biological processes including RNA modification and metabolism (*Figure 2—figure supplement 1A*). We also found that the transcription of several genes encoding proteins that function in animal antibacterial innate immunity was upregulated in response to *P. aeruginosa*, including C-type lectin, glutathione peroxidase, and STING (*Figure 2A and B*). Using an antibody we raised against the C-terminal portion of *M. brevicollis* STING (*Figure 2—figure supplement 1C,D*), we found that STING protein levels are also elevated in response to *P. aeruginosa* (*Figure 2C*). Given the importance of STING in animal immunity and its upregulation in response to *P. aeruginosa*, we pursued its functional relevance in the *M. brevicollis* pathogen response.

## The cyclic dinucleotide 2'3' cGAMP induces elevated expression of STING in *M. brevicollis*

The predicted domain architecture of *M. brevicollis* STING consists of four transmembrane domains followed by a cytosolic STING domain (*Figure 3A*, *Figure 2—figure supplement 1*), and likely matches the structure of the ancestral animal STING protein. Vertebrate STING proteins contain a C-terminal tail (CTT; *Figure 3A*, *Figure 2—figure supplement 1*) that is required for the induction of interferons (*Tanaka and Chen, 2012*; *Liu et al., 2015*; *Zhang et al., 2019*), and for the activation of other downstream responses, including NFkB (*Abe and Barber, 2014*) and autophagy (*Gui et al., 2019*; *Yamashiro et al., 2020*; *Prabakaran et al., 2018*) pathways. Both the STING CTT and interferons evolved in vertebrates, and it is currently unclear how choanoflagellate and invertebrate STING proteins mediate downstream immune responses (*Margolis et al., 2017*; *de Oliveira Mann et al., 2019*). However, the conservation of putative cyclic dinucleotide-binding residues in *M. brevicollis* STING (*Figure 3B*) led us to hypothesize that STING signaling may be induced by cyclic dinucleotides. In addition, because *M. brevicollis* has a cGAS-like enzyme (*Figure 1—figure supplement 1A*), it is possible that *M. brevicollis* produces an endogenous cyclic dinucleotide similar to mammalian 2'3' cGAMP (*Wu et al., 2014*; *Ahn and Barber, 2019*; *Watson et al., 2015*).

To identify potential STING inducers (*Watson et al., 2015*; *Booth et al., 2018*), we treated *M. brevicollis* with purified cyclic dinucleotides, including mammalian cGAMP (2'3' cGAMP) and 3'3'-linked bacterial cyclic dinucleotides (3'3' c-di-GMP, 3'3' c-di-AMP, 3'3' cGAMP). We first performed dose-response curves to determine if the different cyclic dinucleotides affect the viability of *M.*

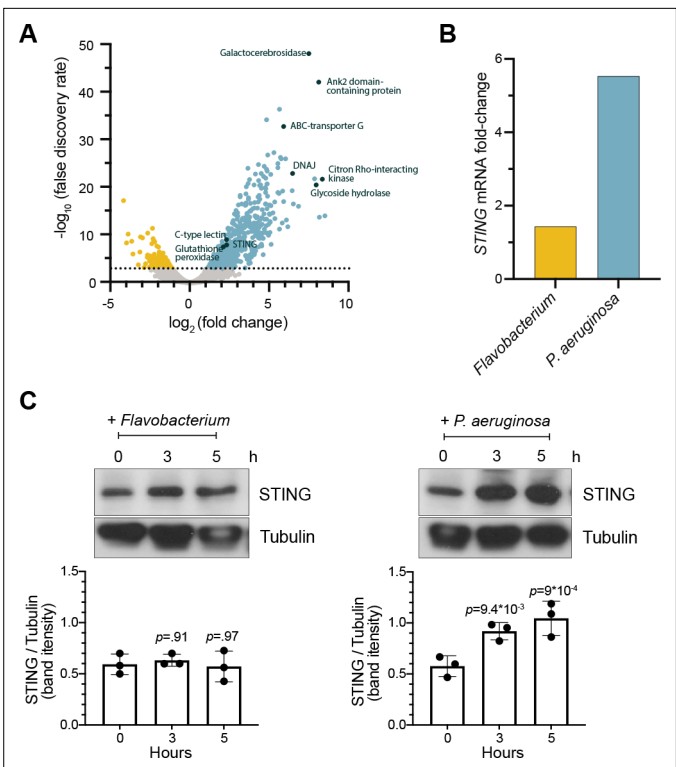

**Figure 2.** STING is upregulated in *M. brevicollis* after exposure to *P. aeruginosa* (**A,B**) *STING* transcript levels increase in response to *P. aeruginosa*. (**A**) Volcano plot displaying genes differentially expressed between *M. brevicollis* exposed to *P. aeruginosa* PAO1 and *Flavobacterium* (control) conditioned medium for three hours. Differentially expressed genes are depicted by blue (674 upregulated genes) and yellow (232 downregulated genes) dots (fold change ≥2; FDR ≤ 1e$^{-4}$). Select genes that are upregulated or may function in innate immunity are labeled. RNA-seq libraries were prepared from four biological replicates. (**B**) After a 3-hr treatment, *STING* mRNA levels (determined by RNA-seq) increase 1.42-fold in cells exposed to *Flavobacterium* conditioned medium and 5.54 fold in cells exposed to *P. aeruginosa* conditioned medium, compared to untreated controls. (**C**) STING protein levels increase after exposure to *P. aeruginosa*. STING levels were examined by immunoblotting at indicated timepoints after exposure to *Flavobacterium* or *P. aeruginosa* conditioned medium (5% vol/vol). Tubulin is shown as loading control, and intensity of STING protein bands were quantified relative to tubulin. Statistical analysis (one-way ANOVA, Dunnett's multiple comparison) was performed in GraphPad software, and p-values shown are calculated using 0 hr timepoint as the control group.

The online version of this article includes the following figure supplement(s) for figure 2:

**Figure supplement 1.** *M. brevicollis* response to *P. aeruginosa*, andSTING antibody validation and protein alignment.

*brevicollis* (*Figure 3C*). Interestingly, we found that exposure to 2′3′ cGAMP-induced cell death in a dose-dependent manner. In contrast, exposure to 3′3′-linked CDNs cGAMP, c-di-GMP, and c-di-AMP did not alter *M. brevicollis* survival. Transcriptional profiling of *M. brevicollis* revealed a robust transcriptional response to 2′3′ cGAMP after three hours (*Figure 3—figure supplement 1A*). Moreover, transcriptional profiling of *M. brevicollis* exposed to 2′3′ cGAMP or 3′3′ cGAMP for 3 hr revealed that *STING* mRNA levels increase in response to 2′3′ cGAMP, but remain unchanged in response to 3′3 cGAMP (*Figure 3—figure supplement 1A-C*). Therefore, we next treated *M. brevicollis* with the cyclic dinucleotides for 5 hr, and measured STING protein levels by immunoblot (*Figure 3D*). Treatment with 2′3′ cGAMP, but not the bacterially produced cyclic dinucleotides, led to elevated levels of STING protein compared to unstimulated cells. A time course of 2′3′ cGAMP treatment revealed that STING protein levels increase as early as 3 hr after exposure to the cyclic dinucleotide and remain elevated for at least 7 hr, approximately one cell cycle (*Figure 3E*). While we also observed sustained upregulation of STING in the presence of *P. aeruginosa*, this is markedly different from what has been described in mammals, wherein STING activation results in its translocation to lysosomes and

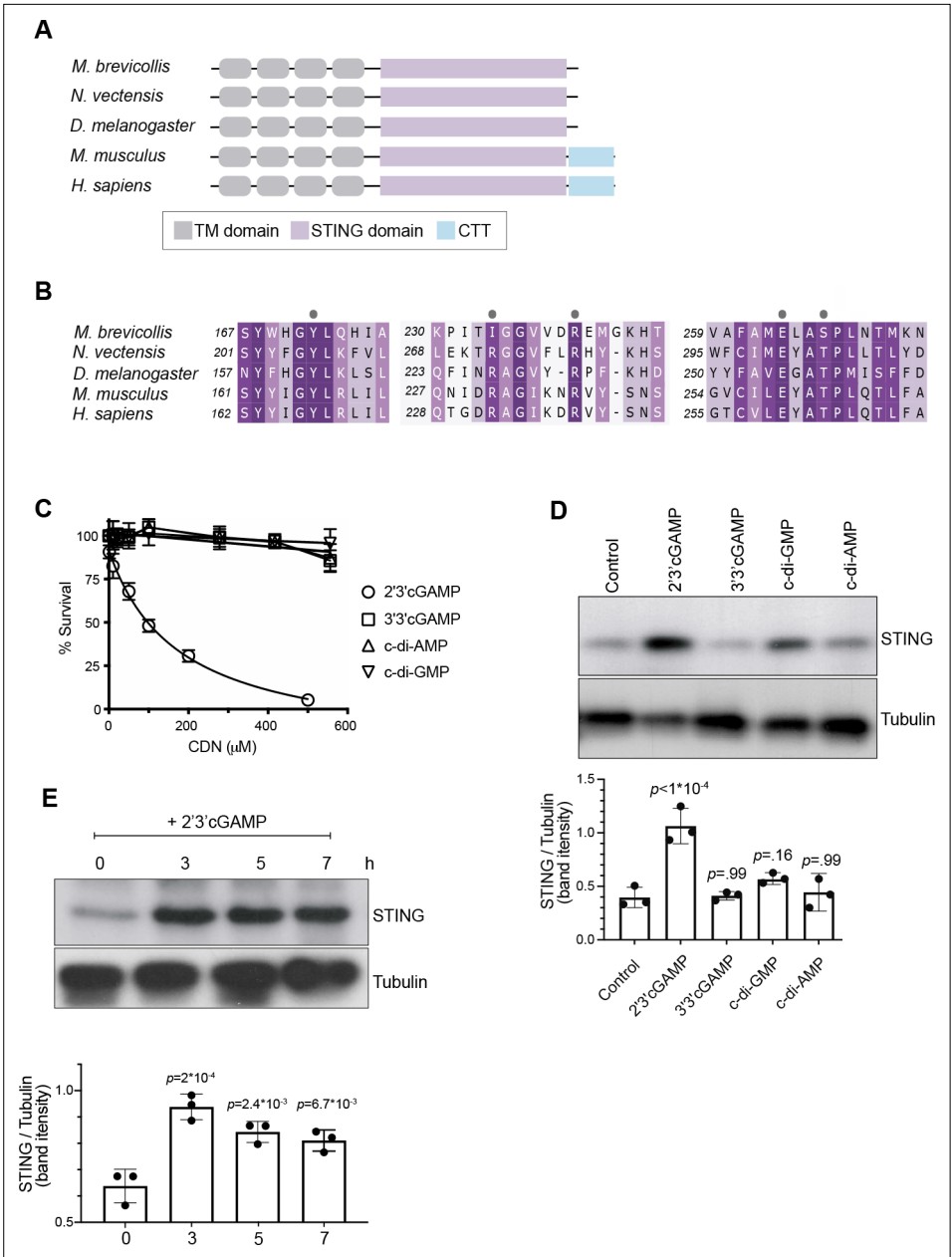

**Figure 3.** 2'3' cGAMP, but not bacterially produced cyclic dinucleotides, induces elevated levels of STING.
(**A**) Schematic of choanoflagellate (*M. brevicollis*), sea anemone (*N. vectensis*), insect (*D. melanogaster*), and mammalian (*M. musculus* and *H. sapiens*) STING proteins. Transmembrane (TM) domains are depicted in gray, STING cyclic dinucleotide binding domain (CDN) in purple, and C-terminal tail domain (CTT) in blue. (**B**) Partial protein sequence alignment (generated by Clustal Omega multiple sequence alignment) of *M. brevicollis* and animal STING proteins, colored by similarity. *M. brevicollis* STING and human STING are 19.1 % identical and 36.6 % similar at the amino acid level. Key cyclic dinucleotide-interacting residues from human STING structure are indicated by circles. (**C**) Dose-response curves of *M. brevicollis* exposed to cyclic dinucleotides for 24 hr reveal that treatment with 2'3'cGAMP, but not 3'3' cGAMP, c-di-AMP, or c-di-GMP, leads to *M. brevicollis* cell death in a dose-dependent manner. Data represent mean ± SD for at least three biological replicates. (**D**) STING protein levels increase after exposure to 2'3'cGAMP, but not bacterially produced cyclic dinucleotides. *M. brevicollis* STING levels were examined by immunoblotting 5 hr after exposure to 2'3'cGAMP (100 µM), 3'3'cGAMP (200 µM), c-di-GMP (200 µM), or c-di-AMP (200 µM). Tubulin is shown as loading control, and intensity of STING protein bands were quantified relative to tubulin. Shown is a representative blot from three biological replicates. Statistical analysis (one-way ANOVA, Dunnett's multiple comparison) was performed in GraphPad software. (**E**) STING protein levels increase and remain elevated after exposure to 100 µM 2'3'cGAMP. Tubulin is shown as loading control,

*Figure 3 continued on next page*

*Figure 3 continued*

and data are representative of three biological replicates. Statistical analysis (one-way ANOVA, Dunnett's multiple comparison) was performed in GraphPad software, and p-values shown are calculated using 0 hr timepoint as control group.

The online version of this article includes the following figure supplement(s) for figure 3:

**Figure supplement 1.** *M. brevicollis* has distinct responses to 2'3' cGAMP and 3'3' cGAMP.

degradation (*Prabakaran et al., 2018*). In addition, immunostaining for STING in fixed *M. brevicollis* revealed that the number and intensity of STING puncta increases after exposure to 2'3' cGAMP (*Figure 3—figure supplement 1E,F*), although the localization of STING was difficult to assess by immunostaining due to a lack of available subcellular markers. These data suggest that *M. brevicollis* STING responds to 2'3' cGAMP, and that this cyclic dinucleotide can be used to further characterize the role of STING in *M. brevicollis*.

## Transfection reveals that STING localizes to the *M. brevicollis* endoplasmic reticulum

A key barrier to investigating gene function in *M. brevicollis* has been the absence of transfection and reverse genetics. We found that the transfection protocol recently developed for the choanoflagellate *Salpingoeca rosetta* (*Booth et al., 2018*) was not effective in *M. brevicollis*, but by implementing a number of alterations to optimize reagents and conditions (see Materials and methods) we were able to achieve both reproducible transfection and establishment of stably transformed cell lines in *M. brevicollis*.

To investigate the subcellular localization of STING, we established a robust transfection protocol for *M. brevicollis* that would allow the expression of fluorescently labeled STING along with fluorescent subcellular markers for different organelles.

We observed that STING-mTFP protein localized to tubule-like structures around the nucleus (*Figure 4A*) similar to what was observed by immunostaining with an antibody to STING (*Figure 3—figure supplement 1E*,F). We then co-transfected STING-mTFP alongside fluorescent reporters marking the endoplasmic reticulum (ER) or mitochondria (*Figure 4B,C*) and performed live-cell imaging. STING-mTFP co-localized with a fluorescent marker highlighting the ER (*Figure 4B*). Thus, as in mammalian cells (*Ishikawa and Barber, 2008*; *Dobbs et al., 2015*), STING localizes to regions of the ER in *M. brevicollis*.

## Genetic disruption of STING reveals its role in responding to 2'3' cGAMP and *P. aeruginosa*

Disrupting the *STING* locus using CRISPR/Cas9-mediated genome editing (*Figure 5A*) enabled us to investigate the function of STING. To overcome low gene editing efficiencies in *M. brevicollis*, we based our gene editing strategy on a protocol recently developed for *S. rosetta* that simultaneously edits a gene of interest and confers cycloheximide resistance (*Booth and King, 2020*). By selecting for cycloheximide resistance and then performing clonal isolation, we were able to isolate a clonal cell line that has a deletion within the *STING* locus that introduces premature stop codons (*Figure 5—figure supplement 1A*). We were unable to detect STING protein in *STING*⁻ cells by immunoblot (*Figure 5B*). Wild type and *STING*⁻ cells have similar growth kinetics (*Figure 5—figure supplement 1B*), suggesting that STING is not required for cell viability under standard laboratory conditions. In addition, overexpression of STING-mTFP did not affect *M. brevicollis* viability.

To investigate the connection between 2'3' cGAMP and STING signaling in *M. brevicollis*, we exposed *STING*⁻ cells to increasing concentrations of 2'3' cGAMP. In contrast to wild-type *M. brevicollis*, *STING*⁻ cells are resistant to 2'3' cGAMP-induced cell death (*Figure 5C*). The 2'3' cGAMP resistance phenotype could be partially reversed by stably expressing STING within the *STING*⁻ mutant background (*Figure 5D*). In addition, *STING*⁻ cells fail to induce a strong transcriptional response to 2'3' cGAMP compared to wild-type cells (*Figure 5E*, *Figure 3—figure supplement 1A*, *Figure 5—figure supplement 1C,D*). While 371 genes are differentially expressed in wild-type cells after exposure to 2'3' cGAMP for 3 hr, only 28 genes are differentially expressed in *STING*⁻ cells (FC ≥3; FDR ≤ $10^{-4}$). Thus, 2'3' cGAMP induces a STING-dependent transcriptional response in *M. brevicollis*.

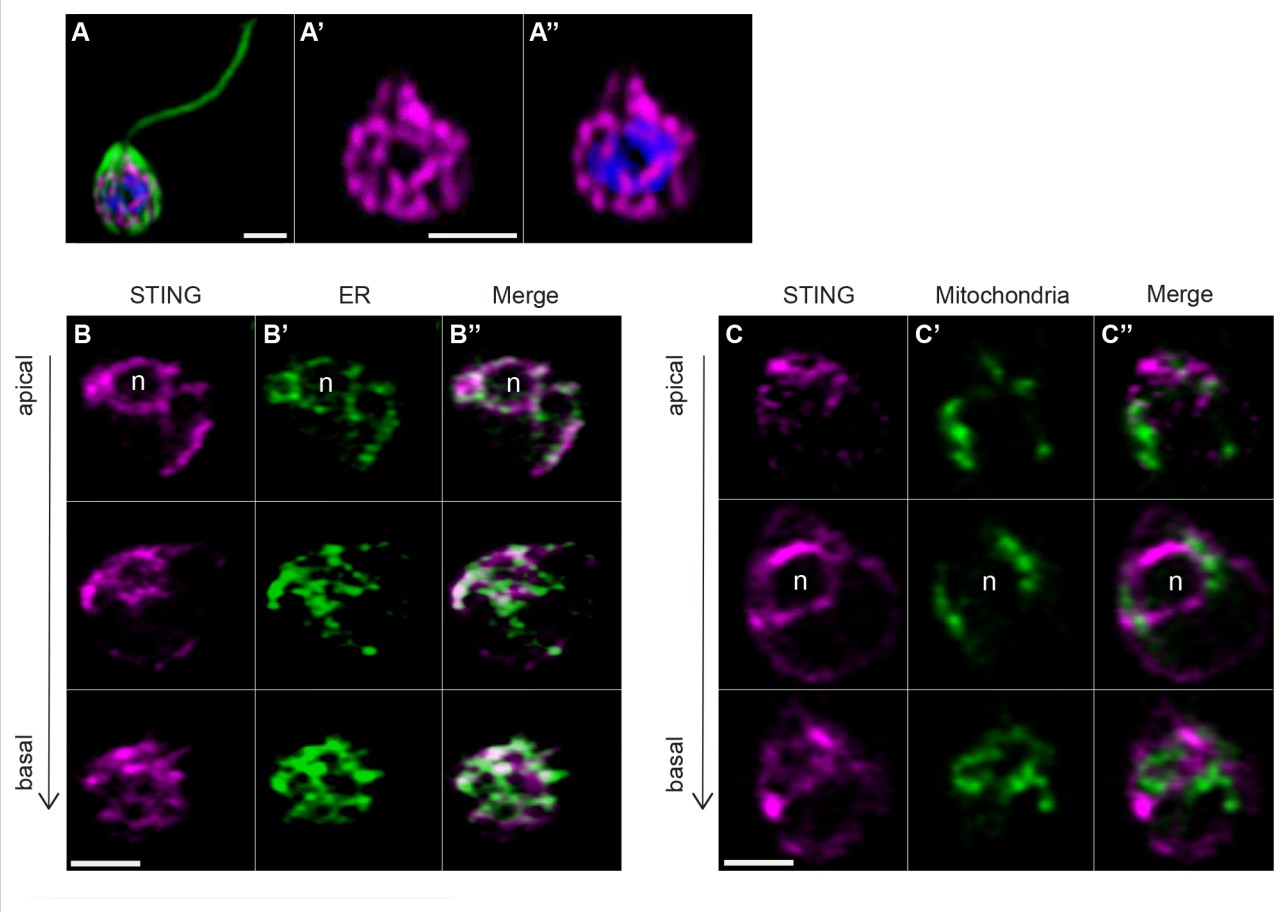

**Figure 4.** Transfection reveals STING localization to perinuclear and endoplasmic reticulum regions. (**A**) STING-mTFP localizes to tubule-like structures around the nucleus in cells stably expressing STING-mTFP. Green: anti-tubulin antibody (flagella and cell body), magenta: anti-STING antibody, blue: Hoechst. Scale bar represents 2 μm. (**B,C**) Fluorescent markers and live cell imaging reveal that STING is localized to the endoplasmic reticulum (ER). Cells were co-transfected with STING-mTFP and an mCherry fusion protein that localizes either to the endoplasmic reticulum (B; mCherry-HDEL) or mitochondria (C; Cox4-mCherry) (*King et al., 2008*). Cells were recovered in the presence of *Flavobacterium* feeding bacteria for 28 hr after co-transfection, and then live cells were visualized with super-resolution microscopy. Each panel shows Z-slice images of a single representative cell. In confocal Z-slice images, cells are oriented with the apical flagella pointing up, and the nucleus is marked by 'n' when clearly included in the plane of focus. STING colocalized with the ER marker (**B"**), but not the mitochondrial marker (**C"**). Scale bar represents 2 μm.

Interestingly, of the 22 choanoflagellate species with sequenced transcriptomes (*Richter et al., 2018*; *Brunet et al., 2019*), only *M. brevicollis* and *Salpingoeca macrocollata*, express homologs of both STING and cGAS (*Figure 1—figure supplement 1A*, *Figure 5F*, *Figure 5—figure supplement 1E*). Therefore, we were curious whether other choanoflagellate species are able to respond to 2'3' cGAMP in the absence of a putative STING protein. We exposed four other choanoflagellate species (*Salpingoeca infusionum*, *S. macrocollata*, *S. rosetta*, and *Salpingoeca punica*) to increasing 2'3' cGAMP concentrations, and quantified survival after 24 hours (*Figure 5G*). Of these additional species, only *S. macrocollata* had impaired survival in the presence of 2'3' cGAMP. Thus, it is possible that STING also responds to 2'3' cGAMP in *S. macrocollata*.

We next asked whether *STING⁻* cells have altered responses to other immune agonists. Although *M. brevicollis* is continuously co-cultured with feeding bacteria, we observed that treatment with high concentrations of *E. coli* lipopolysaccharides induces cell death (*Figure 5H*). As LPS is not known to activate STING signaling, we treated wild type and *STING⁻* cells with LPS to probe the specificity of STING-mediated immune responses in *M. brevicollis*. The survival responses of wild type and *STING⁻* cells to LPS were indistinguishable (*Figure 5H*), suggesting that there are separable pathways for responding to 2'3' cGAMP and LPS. We also examined the survival of *STING⁻* cells exposed to *P. aeruginosa* conditioned medium (*Figure 5I and J*). In growth curve experiments, *P. aeruginosa*

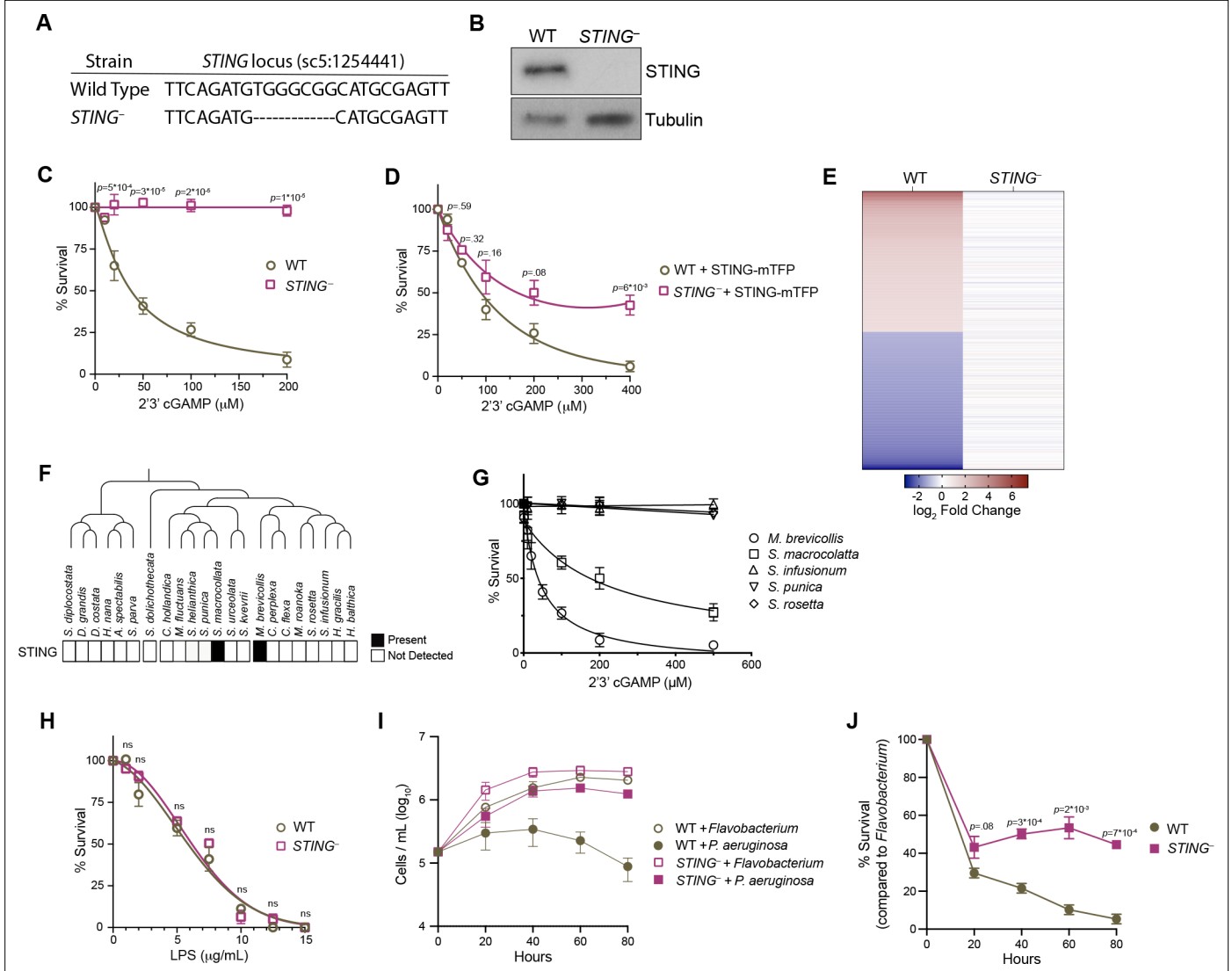

**Figure 5.** STING mediates responses to 2'3'cGAMP and *P. aeruginosa*. (**A**) The genotypes of wild type and genome-edited *STING⁻* strains at the *STING* locus. (**B**) STING protein is not detectable by immunoblot in *STING⁻* cells. Shown is a representative blot from three biological replicates. (**C,D**) STING is necessary for 2'3'cGAMP-induced cell death. (**C**) Wild type and *STING⁻* strains were treated with increasing concentrations of 2'3'cGAMP, and survival was quantified after 24 hr. In contrast to wild type cells, 2'3'cGAMP does not induce cell death in *STING⁻* cells. Data represent mean ± SD for four biological replicates. (**D**) Wild type and *STING⁻* cells were transfected with STING-mTFP, and treated with puromycin to generate stable clonal strains. Stable expression of STING-mTFP in *STING⁻* cells partially rescued the phenotype of 2'3'cGAMP-induced cell death. Data represent mean ± SD for three biological replicates. Statistical analysis (multiple unpaired t-tests) was performed in GraphPad software. (**E**) Wild type and *STING⁻* strains have distinct transcriptional responses to 2'3' cGAMP. Differential expression analysis was performed on wild type and *STING⁻* cells treated with 100 µM 2'3'cGAMP or a vehicle control for 3 hr. A heatmap comparing the log₂ fold change of genes identified as differentially expressed (FC ≥2; FDR ≤ 10⁻⁴) in wild-type cells after 2'3' cGAMP treatment, to their log₂ fold change in *STING⁻* cells after 2'3' cGAMP treatment. RNA-seq libraries were prepared from two biological replicates. (**F**) Presence of STING in the transcriptomes of diverse choanoflagellate species. Data from *Richter et al., 2018*. (**G**) Effects of 2'3'cGAMP on different choanoflagellate species. Choanoflagellates were grown to late-log phase, and treated with increasing concentrations of 2'3'cGAMP. Survival was quantified after 24 hr. 2'3'cGAMP only affected the survival of *M. brevicollis* and *S. macrocollata*, the two sequenced choanoflagellate species with a STING homolog. Data represent mean ± SD for three biological replicates. (**H**) Wild type and *STING⁻* cells have similar survival responses to LPS, suggesting that STING is not required for mediating a response to LPS. Wild type and *STING⁻* strains were treated with increasing concentrations of *E. coli* LPS, and survival was quantified after 24 hr. Data represent mean ± SD for four biological replicates. Statistical analysis (multiple unpaired t-tests) was performed in GraphPad software. (**I,J**) STING renders *M. brevicollis* more susceptible to *P. aeruginosa*-induced growth inhibition. (**I**) Wild type and *STING⁻* cells were exposed to control *Flavobacterium* or *P. aeruginosa* conditioned medium (5% vol/vol), and cell densities were quantified at indicated time points. Data represent mean ± SD for three biological replicates. (**J**) Percent survival calculated from growth curves in (**I**). Statistical analysis (multiple unpaired t-tests) was performed in GraphPad software.

*Figure 5 continued on next page*

*Figure 5 continued*

The online version of this article includes the following figure supplement(s) for figure 5:

**Figure supplement 1.** Characterizing *STING⁻ M. brevicollis*.

---

hindered the growth rate and stationary phase cell density of *STING⁻* cells compared to *Flavobacterium* (*Figure 5I and J*). However, *STING⁻* cells were still able to divide in the presence of *P. aeruginosa*, whereas wild-type cell growth was completely restricted (*Figure 5I and J*). These results indicate that wild-type cells are more susceptible to *P. aeruginosa* than *STING⁻* cells, although it is unclear how STING contributes to *P. aeruginosa*-induced growth restriction and cell death.

## 2'3' cGAMP-induces autophagic signaling via STING

One downstream consequence of STING signaling in animals is the initiation of autophagy (*Moretti et al., 2017*; *Gui et al., 2019*; *Yamashiro et al., 2020*; *Watson et al., 2015*; *Liu et al., 2018*). Based on viral infection studies in *D. melanogaster* (*Liu et al., 2018*) and experiments expressing invertebrate STING in mammalian cells (*Gui et al., 2019*), it has been proposed that the induction of autophagy may be an interferon-independent ancestral function of STING. Although *M. brevicollis* lacks many effectors required for immune responses downstream of STING in animals (including TBK1 and NF-kB; Figure S1A), autophagy machinery is well conserved in *M. brevicollis*. Therefore, we asked if one outcome of 2'3' cGAMP exposure in *M. brevicollis* is the induction of autophagy.

The evolutionarily conserved protein Atg8/LC3 is a ubiquitin-like protein that can be used to monitor autophagy (*Klionsky, 2021*; *Tanida et al., 2005*). During autophagosome formation, unmodified Atg8, called Atg8-I, is conjugated to phosphatidylethanolamine. Lipidated Atg8, called Atg8-II, remains associated with growing autophagosomes. As such, two indicators of autophagy are elevated Atg8-II levels relative to Atg8-I and increased formation of Atg8+ autophagosome puncta. Because antibodies are not available to detect endogenous *M. brevicollis* autophagy markers or cargo receptors, we generated wild type and *STING⁻* cell lines stably expressing mCherry-Atg8. Stable expression of mCherry-Atg8 under the control of the constitutive pEFL promoter did not alter the relative susceptibilities of these cell lines to 2'3'cGAMP (*Figure 6—figure supplement 1A*). By immunoblot, mCherry-Atg8-II can be distinguished from mCherry-Atg8-I based on its enhanced gel mobility. When we exposed both cell lines to 2'3' cGAMP for three hours, we observed increased levels of Atg8-II relative to Atg8-I by immunoblot in wild type, but not *STING⁻* cells (*Figure 6A*). These results suggest that treatment with 2'3' cGAMP induces autophagic signaling in a STING-dependent manner; however, making this conclusion requires evidence of autophagy induction through inhibitor studies. To confirm autophagy induction, we treated cells with chloroquine, a lysosomotropic agent which inhibits autophagy by blocking endosomal acidification, thereby preventing amphisome formation and Atg8-II turnover (*Klionsky, 2021*). Exposing wild-type cells pretreated with chloroquine to 2'3' cGAMP for 3 hr resulted in increased levels of Atg8-II relative to Atg8-I, suggesting that 2'3' cGAMP treatment indeed induces the autophagic pathway (*Figure 6B*, *Figure 6—figure supplement 1B*). In cells pretreated with chloroquine, STING levels did not markedly increase after exposure to 2'3' cGAMP (*Figure 6—figure supplement 1B*), raising the possibility that the autophagic pathway is important for regulating STING protein levels. We next examined whether 2'3' cGAMP induces Atg8+ puncta formation by treating wild type and *STING⁻* cells with 2'3' cGAMP for 3 hr, and observing mCherry foci by microscopy (*Figure 6C–F*). Quantifying images revealed that Atg8+ puncta accumulate after 2'3' cGAMP treatment in wild type, but not *STING⁻* cells (*Figure 6G*). Overall, these results suggest that *M. brevicollis* responds to 2'3' cGAMP through STING-dependent induction of the autophagy pathway.

Finally, we asked whether STING-mediated autophagic pathway induction affects survival after exposure to 2'3' cGAMP (*Figure 6H*). To determine if inhibiting autophagy impacts 2'3' cGAMP-induced cell death, we examined the survival responses of wild type and *STING⁻ M. brevicollis* to 2'3' cGAMP after pretreatment with lysosomotropic agents chloroquine or NH₄Cl. Chloroquine or NH₄Cl pretreatment rescued 2'3' cGAMP-induced cell death in wild type *M. brevicollis* (*Figure 6H*), whereas the survival of *STING⁻* cells, which are already resistant to 2'3' cGAMP-induced cell death, was not affected. Therefore, we hypothesize that 2'3' cGAMP induces cell death in *M. brevicollis* by overstimulating STING-mediated autophagic signaling.

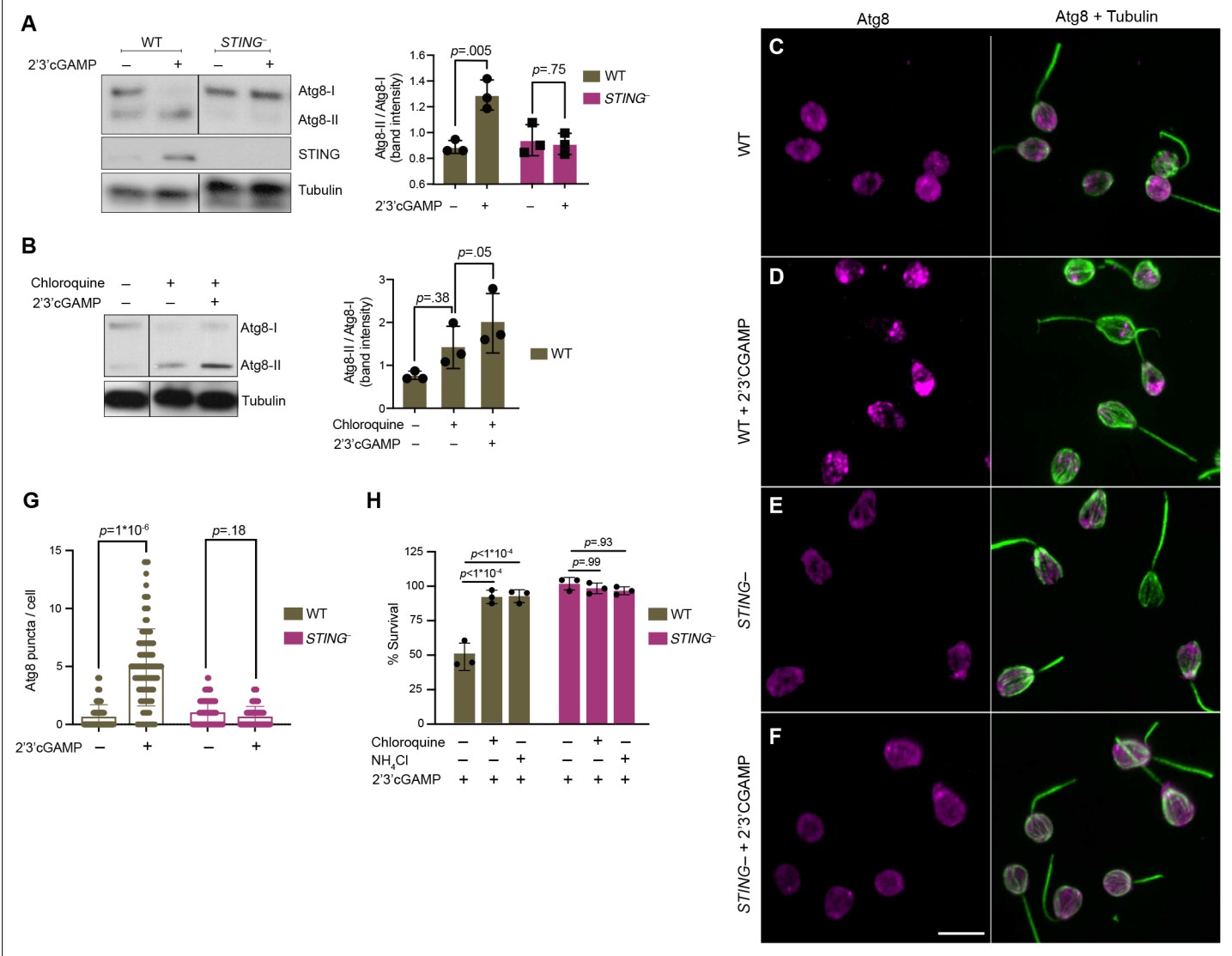

**Figure 6.** STING mediates 2'3'cGAMP-induced autophagic pathway. (**A**) 2'3'cGAMP-induced Atg8 lipidation requires STING. WT and *STING⁻* cells stably expressing mCherry-Atg8 were treated with a vehicle control or 100 μM 2'3'cGAMP for 3 hr, followed by immunoblotting. The band intensity of Atg8-I (unmodified Atg8) and Atg8-II (lipidated Atg8) were quantified for each sample. Relative levels of Atg8 lipidation were assessed by dividing the band intensities of Atg8-II/Atg8-I. Tubulin is shown as loading control. Immunoblot is representative of three biological replicates. (**B**) 2'3'cGAMP induces Atg8 lipidation in chloroquine-treated wild type cells. WT cells stably expressing mCherry-Atg8 were first incubated with 40 mM chloroquine for 6 hr, and then treated with a vehicle control or 100 μM 2'3'cGAMP for 3 hr in the presence of chloroquine, followed by immunoblotting. For each sample, relative levels of Atg8 lipidation were assessed by dividing the band intensities of Atg8-II/Atg8-I. Tubulin is shown as loading control. Immunoblot is representative of three biological replicates. For a representative immunoblot and quantification of Atg8-II/Atg8-I levels in chloroquine-treated *STING⁻* cells, refer to ***Figure 6—figure supplement 1B***. (**C–G**) STING is required for 2'3'cGAMP-induced autophagosome formation. (**C–F**) WT and *STING⁻* cells stably expressing mCherry-Atg8 were treated with a vehicle control or 100 μM 2'3'cGAMP for 3 hr, and then fixed and immunostained. (**C,D**) Representative confocal images of wild-type cells show that Atg8 puncta accumulate after 2'3'cGAMP treatment. Magenta: mCherry-Atg8; Green: anti-tubulin antibody (flagella and cell body). (**E,F**) Representative confocal images of *STING⁻* cells show that Atg8 remains evenly distributed in the cytoplasm after 2'3'cGAMP treatment. (**G**) The number of Atg8 puncta/cell was quantified for WT and *STING⁻* cells treated with a vehicle control or 2'3'cGAMP for 3 hr. Data represent cells quantified from two biological replicates (n = 150 cells per treatment group). Statistical analyses (unpaired two-tailed t-tests) were performed in GraphPad software. (**H**) Treatment with lysosomotropic agents that inhibit autophagy rescue 2'3' cGAMP-induced cell death in wild-type cells. WT and *STING⁻* cells were pre-treated with 40 mM chloroquine, 10 mM NH₄Cl, or a vehicle control for 6 hr. Cells were then exposed to either 100 μM 2'3'cGAMP or a vehicle control for 18 hr before quantifying survival. Data represent mean ± SD for three biological replicates. Statistical analyses (multiple unpaired t-tests) were performed in GraphPad software.

The online version of this article includes the following figure supplement(s) for figure 6:

**Figure supplement 1.** STING mediates 2'3'cGAMP-induced autophagic signaling.

## Discussion

Investigating choanoflagellate immune responses has the potential to inform the ancestry of animal immune pathways. In this study, we screened diverse bacteria to identify a choanoflagellate pathogen, and determined that *M. brevicollis* is killed by sustained exposure to *P. aeruginosa* bacteria. We found that STING, a crucial component of animal innate responses to cytosolic DNA, is upregulated in *M. brevicollis* after exposure to *P. aeruginosa* or the STING ligand 2'3' cGAMP. The application of newly developed transgenic and genetic tools for *M. brevicollis* revealed that, similar to mammalian STING, *M. brevicollis* STING localizes to perinuclear endoplasmic reticulum regions. In addition, STING mediates responses to *P. aeruginosa* bacteria, and is required for inducing transcriptional changes and autophagic signaling in response to 2'3' cGAMP. These data reveal that STING plays conserved roles in choanoflagellate immune responses, and provide insight into the evolution of STING signaling on the animal stem lineage.

The discovery that *M. brevicollis* STING mediates immune responses raises a number of interesting questions about the full extent of its physiological roles in choanoflagellates. For example, while our results demonstrate that *M. brevicollis* STING responds to exogenous 2'3' cGAMP, the endogenous triggers of STING activation in *M. brevicollis* remain to be determined. *M. brevicollis* has a putative cGAS homolog, suggesting that STING may respond to an endogenously produced cyclic dinucleotide similar to 2'3' cGAMP. Determining the enzymatic activities of *M. brevicollis* cGAS and identifying the endogenous trigger of *M. brevicollis* STING will be critical steps towards elucidating mechanisms of STING activation. Although cGAS and STING are rare among sequenced choanoflagellate species, both species with STING homologs, *M. brevicollis* and *S. macrocollata*, also harbor a cGAS homolog (*Figure 1—figure supplement 1A*), suggesting the presence of an intact choanoflagellate cGAS-STING pathway.

Our results suggest that *M. brevicollis* has distinct responses to 2'3' cGAMP versus 3'3'-linked cyclic dinucleotides produced by bacteria (*Figure 3C and D*, *Figure 3—figure supplement 1*). In contrast to 2'3' cGAMP, the purified bacterial cyclic dinucleotides tested (3'3' cGAMP, c-di-AMP, c-di-GMP) do not lead to STING upregulation or induce cell death in *M. brevicollis*. However, 3'3' cGAMP induces a robust transcriptional response in *M. brevicollis*, indicating that STING, or a different cyclic dinucleotide receptor (*McFarland et al., 2017*), responds to these bacterial molecules. One hypothesis is that *M. brevicollis* STING, similar to animal STING proteins (*Kranzusch et al., 2015*), may have different binding affinities for 2'3' and 3'3'-linked cyclic dinucleotides. It is also possible that bacterial cyclic dinucleotides activate additional pathways that influence survival in *M. brevicollis*. As bacterivores, choanoflagellates likely benefit from a fine-tuned response to bacterial cyclic dinucleotides that enables them to interpret higher and lower concentrations in their environment. Elucidating mechanisms of STING activation in *M. brevicollis* could help reveal how STING proteins in animals evolved to respond to both bacterially produced and endogenous cyclic dinucleotides.

While it is clear that 2'3' cGAMP stimulates STING-dependent transcriptional responses and autophagic signaling in *M. brevicollis* (*Figures 5 and 6*), the signaling pathways downstream of STING in choanoflagellates are unknown. Much of what is known about STING signaling comes from mammals and involves the extended CTT domain of STING, which *M. brevicollis* lacks, and immune genes that are restricted to vertebrates. Two pathways downstream of STING activation that are conserved in invertebrates, and as such are proposed ancestral functions of STING, are autophagy and NF-kB signaling. Here, we observed that STING is required for induction of the autophagy pathway in response to 2'3' cGAMP in *M. brevicollis*, indicating that the role of STING in regulating autophagy predates animal origins. While exposure to 2'3' cGAMP leads to NF-kB activation in the sea anemone *N. vectensis* (*Margolis, 2021*) and in insects (*Cai et al., 2020*; *Martin et al., 2018*; *Goto et al., 2018*; *Hua et al., 2018*), neither *M. brevicollis* nor *S. macrocollata*, the two choanoflagellate species with STING, possess a NF-kB homolog (*Richter et al., 2018*). Nonetheless, 2'3' cGAMP activates an extensive transcriptional response downstream of STING in *M. brevicollis*, although the specific signaling pathways remain to be identified. This does not negate the hypothesis that STING signaling led to NF-kB activation in the Urmetazoan, but strongly suggests that additional pathways exist downstream of STING activation in choanoflagellates, and potentially in animals.

Choanoflagellates forage on diverse environmental bacteria for sustenance, yet how they recognize and respond to pathogens is a mystery. Our finding that *P. aeruginosa* has pathogenic effects on *M. brevicollis* (*Figure 1*) provides a much-needed framework for uncovering mechanisms of pathogen

recognition and antibacterial immunity in choanoflagellates. Profiling the host transcriptional response to *P. aeruginosa* has allowed us to identify choanoflagellate genes that may be involved in recognizing (C-type lectins) and combating (polysaccharide lyases, antimicrobial peptides) bacteria; yet, it has also revealed the immense complexity of this interaction, with more than 600 *M. brevicollis* genes differentially expressed in response *P. aeruginosa*. Thus, identifying specific *P. aeruginosa* virulence factors will be critical for understanding why *P. aeruginosa* – but not other bacteria – have pathogenic effects on *M. brevicollis,* and facilitate the characterization of mechanisms underlying choanoflagellate pathogen responses.

With the establishment of molecular genetic techniques in choanoflagellates – first for *S. rosetta* (*Booth et al., 2018*; *Booth and King, 2020),* and here for *M. brevicollis* – we now have the opportunity to explore the functions of candidate immune genes. Identifying additional choanoflagellate pathogens, particularly viral pathogens, will also be key to delineating immune response pathways. Finally, as choanoflagellates are at least as genetically diverse as animals (*Richter et al., 2018*), expanding studies of immune responses to diverse choanoflagellate species will be essential for reconstructing the evolution of immune pathways in animals.

## Materials and methods
### Culturing choanoflagellates

All strains of *M. brevicollis* were co-cultured with *Flavobacterium sp.* bacteria (*King et al., 2008*) (American Type Culture Collection [ATCC], Manassas, VA; Cat. No. PRA-258) in a seawater based media enriched with glycerol, yeast extract, peptone and cereal grass (details in Media Recipes). Cells were grown either at room temperature, or at 16 °C in a wine cooler (Koldfront). All *M. brevicollis* cell lines were verified by 18 S sequencing and RNA-seq. Choanoflagellate cell lines *S. rosetta*, *S. macrocollata*, *S. punica*, and *S. infusionum* were verified by 18 S sequencing.

### Bacterial effects on *M. brevicollis*
#### Isolating environmental bacteria

Environmental bacterial species were isolated from water samples from Woods Hole, MA, St. Petersburg, FL, and Dallas, TX. Water samples were streaked onto Sea Water Complete media or LB plates, and grown at 30 ° C or 37 ° C. After isolating individual colonies, partial 16 S sequencing using 16 S universal primers (27 F: 5′-AGAGTTTGATCCTGGCTCAG-3′, 1492 R: 5′-TACGGYTACCTTGTTA CGACTT-3′) was used to determine the identity of the bacterial isolates.

#### Screening for pathogenic effects

*M. brevicollis* was grown for 30 hr, and feeding bacteria were reduced through one round of centrifugation and resuspension in artificial seawater (ASW). Cells were counted on a hemocytometer and diluted to $5 \times 10^6$ cells/mL in High Nutrient Medium, and plated into 24-well plates.

For each bacterium, a single colony was inoculated into LB and grown shaking overnight at either 30 ° C (environmental isolates) or 37 ° C (mouse isolates). Bacterial cells were pelleted by centrifugation for 5 minutes at 4000 x *g,* and resuspended in artificial seawater (ASW) to an OD~1.

Each bacterial species was added to *M. brevicollis* culture at two concentrations (10 mL/mL and 50 mL/mL) in duplicate. *M. brevicollis* was then monitored at regular intervals for changes in behavior and growth.

#### Growth curves in the presence of bacteria

All bacteria were grown shaking at 30 ° C in Sea Water Complete media or LB (to optical density of 0.8). For each bacterial strain, CFU plating was used to estimate the number of bacterial cells/ mL under these growth conditions. To prepare bacterial conditioned media, bacterial cells were pelleted by centrifugation for 10 minutes at 4000 x *g,* and supernatant was passed through a 0.22 mm sterilizing filter.

*M. brevicollis* was grown for 30 hr, and bacteria were washed away through two consecutive rounds of centrifugation and resuspension in artificial seawater (ASW). Cells were counted on a hemocytometer and diluted to $1.0 \times 10^6$ cells/mL (growth curves with live bacteria) or $1.5 \times 10^5$ cells/mL (growth curves with conditioned medium) in High Nutrient Medium. To test the effects of live bacteria, 1.5 ×

10⁶ bacterial cells were added per 1 mL of *M. brevicollis* culture. To test the effects of bacterial conditioned media, 50 ml of bacterial conditioned media was added per 1 mL of *M. brevicollis* culture. For each growth curve biological replicate, cells were plated into 24-well plates, and two wells were counted per time point as technical replicates. At least three biological replicates are represented in each graph.

## Bacterial internalization

Fluorescent *E. coli* and *P. aeruginosa* were grown shaking at 30 ° C in LB to an optical density of OD$_{600}$ = 0.8. Fluorescent *C. jejuni* was grown from freezer stocks in microaerobic conditions on Mueller-Hinton agar. For each bacterial strain, CFU plating was used to estimate the number of bacterial cells/ mL under these growth conditions.

*M. brevicollis* was grown for 30 hr, and feeding bacteria were washed away with one round of centrifugation and resuspension in artificial seawater (ASW). Cells were counted and diluted to 1.5 × 10⁵ cells/mL in ASW. 1.5 × 10⁷ bacterial cells were added to 2 mL *M. brevicollis* culture (MOI = 50), and co-incubated at room temperature for 1 hr with gentle mixing at regular intervals to avoid settling. To quantify bead internalization, *M. brevicollis* was co-incubated with bacteria for 1 hr (as described above), at which point ~1 × 10¹⁰ beads (0.2 mm diameter, resuspended in 1 % BSA to prevent clumping) were added to the conical for an additional 30 min.

Prior to imaging, 200 mL aliquots were transferred to eight-well glass bottom chambers (Ibidi Cat. No 80827). Live imaging was performed on a Zeiss Axio Observer widefield microscope using a 63 x objective. Images were processed and analyzed using Fij (*Schindelin et al., 2012*).

## *P. aeruginosa* deletion mutants

*P. aeruginosa* deletion strains were acquired from the Seattle PAO1 transposon mutant library (NIH P30 DK089507). Strains RP436 and RP576 (PMID 15731071) were acquired from Russell Vance. The effects of both live bacteria and bacterial conditioned medium were tested for all acquired strains at a range of PFU/mL (live bacteria) or percent volume (conditioned medium).

## Conditioned media treatments

*P. aeruginosa* was grown shaking at 30 ° C in LB for 24 hr, and bacterial conditioned media was prepared by pelleting bacterial cells by centrifugation for 10 min at 4000 x *g,* and filtering supernatant twice through 0.22 mm sterilizing filters. To determine if the bioactivity of the conditioned media is present in secreted outer membrane vesicles (OMVs), OMVs were depleted from the conditioned media by ultracentrifugation as described in *Woznica et al., 2016*. To determine if the bioactivity of the conditioned media is heat labile, conditioned media was incubated at 100 ° C for 20 min. To determine if the bioactivity of the conditioned media is due to a protein, conditioned media was incubated with 200 μg/mL proteinase K (New England Biolabs) at 37 ° C for 12 hr, after which the enzyme was deactivated by heating the conditioned media to 80 ° C for 30 min. To determine if the bioactivity of the conditioned media is due to a nucleic acid, the conditioned media boiled for 20 min at 100 ° C, lyophilized, and resuspended in 40 mM Tris-Cl (pH = 8, 10 mM MgCl$_2$, 1 mM CaCl$_2$) for digestion with RNase A (100 μg mL$^{-1}$; Sigma), DNase I (100 μg mL$^{-1}$; Sigma), or Phosphodiesterase I from *Crotalus adamanteus* venom (2.5mU; Sigma). For digestion with Nuclease P1 from *Penicillium citrinum* (2.5mU; Sigma), lyophilized conditioned media was resuspended in 40 mM Tris-Cl (pH = 6, 2 mM ZnCl$_2$). All nucleic acid digestions took place for 2.5 hr at 37 °C. To test whether the bioactivity of the conditioned media is in the methanolic extract, conditioned media was lyophilized and resuspended in methanol. The methanolic suspension was centrifuged at 8000 rpm for 5 min, and the methanol layer recovered and dried.

## Immune agonist dose-response curves

*M. brevicollis* was grown to late-log phase, and feeding bacteria were reduced through one round of centrifugation and resuspension in artificial seawater (ASW). Cells were counted on a hemocytometer and diluted to 1.0 × 10⁶ cells/mL (growth curves with live bacteria) in High Nutrient Medium, and aliquoted into 96-well (100 μL/well) or 24-well (1 mL/well) plates. Immune agonists were added at indicated concentrations in technical duplicate, and cells were counted again after 24 hr. % survival is

a calculation of: [mean experimental (cells/mL) / mean control (cells/mL)]. Each dose-response curve is representative of at least three biological replicates.

## RNA-Seq

### Growth of choanoflagellate cultures

After thawing new cultures, growth curves were conducted to determine the seeding density and time required to harvest cells at late-log phase growth. To grow large numbers of cells for RNA-seq, cells were seeded one to two days prior to the experiment in either three-layer flasks (Falcon; Corning, Oneonta, NY, USA; Cat. No. 14-826-95) or 75 cm$^2$ flasks (Falcon; Corning, Oneonta, NY, USA; Cat. No. 13-680-65), and grown at room temperature. Bacteria were washed away from choanoflagellate cells through two rounds of centrifugation and resuspension in artificial seawater (ASW). To count the cell density, cells were diluted 100-fold in 200 µl of ASW, and fixed with 1 µL of 16 % paraformaldehyde. Cells were counted on a hemocytometer, and the remaining cells were diluted to a final concentration of 4 × 10$^6$ choanoflagellate cells/mL. The resuspended cells were divided into 2.5 mL aliquots and plated in six-well plates prior to treatment. After treatment, cells were transferred to a 15 mL conical and pelleted by centrifugation at 2400 x $g$ for 5 min, flash frozen with liquid nitrogen, and stored at –80 °C.

### RNA isolation

Total RNA was isolated from cell pellets with the RNAqueous kit (Ambion, Thermo Fisher Scientific). Double the amount of lysis buffer was used to increase RNA yield and decrease degradation, and RNA was eluted in minimal volumes in each of the two elution steps (40 µL and 15 µL). RNA was precipitated in LiCl to remove contaminating genomic DNA. Total RNA concentration and quality was evaluated using the Agilent Bioanalyzer 2,100 system and RNA Nano Chip kit (Cat No. 5067–1511).

### Library preparation, sequencing, and analysis

Libraries were prepared and sequenced by the UTSW Genomics Sequencing Core. RNA libraries were generated with the Illumina TruSeq Stranded mRNA Library prep kit (Cat No. 20020594), using a starting total RNA input of 2–3 µg. To remove contaminating bacterial RNA, samples were first poly-A selected using oligo-dT attached magnetic beads. Following purification, the mRNA was fragmented at 94 °C for 4 min, and cleaved RNA fragments were synthesized into cDNA. After an end repair step, UMI adapters (synthesized by IDT) were ligated to the cDNA, and the products were twice purified using AMPure XP beads before amplification.

Library quantity was measured using the Quant-iT PicoGreen dsDNA Assay kit by Invitrogen (Cat No. P7589) and a PerkinElmer Victor X3, 2030 Multilabel Reader. Library quality was verified on an Agilent 2,100 Bioanalyzer instrument using Agilent High sensitivity DNA kit (Cat No. 5067–4626) or DNA 1000 kit (Cat No. 5067–1504). Libraries were pooled, and sequenced in different batches on either the Illumina NextSeq 550 system with SE-75 workflow, or the Illumina NovaSeq 6,000 system with S4 flowcell and XP PE-100 workflow, generating 25–40 million reads per sample. Reads were checked for quality using fastqc (v0.11.2) and fastq_screen (v0.4.4), and trimmed using fastq-mcf (ea-utils, v1.1.2–806). Trimmed fastq files were mapped to the *Monosiga brevicollis* reference genome (NCBI:txid81824) using TopHat (**Kim et al., 2013**) (v2.0.12). Duplicates were marked using picard-tools (v2.10.10). Read counts were generated using featureCounts (**Liao et al., 2014**), and differential expression analysis was performed using edgeR (**Robinson et al., 2010**). Statistical cutoffs of FDR ≤ 10$^{-4}$ were used to identify significant differentially expressed genes. GO enrichment analysis of differentially expressed genes was performed using DAVID (https://david.ncifcrf.gov/).

### RT-qPCR

RNA was isolated as described above, and cDNA was synthesized from total RNA with the High Capacity cDNA Reverse Transcription kit (Applied Biosystems; Thermo Fisher). Real-time PCR was performed with iTaq Universal SYBR Green Supermix (Biorad; Cat No. 1725121) or SYBR Green PCR master mix (Applied Biosystems; Cat No. 4309155) in either an 7,500 Fast Real-Time PCR System (Applied Biosystems), or a QuantStudio 3 Real-Time PCR System (Applied Biosystems). Ct values were converted into relative gene expression using the ΔΔCt method (**Livak and Schmittgen, 2001**) and the internal control gene RPL15 (MONBRDRAFT_38309).

## Immunoblotting

*M. brevicollis* was harvested by centrifugation at 5000 x *g* for 5 min at 4 °C, and resuspended in 100 µL lysis buffer (50 mM Tris, pH 7.4, 150 mM NaCl, 1 mM EDTA, 1 mM ethyleneglycoltetraacetic acid [EGTA], 0.5 % sodium deoxycholate, 1% NP-40) containing protease inhibitor cocktail (Roche) for 10 min at 4 °C. The crude lysate was clarified by centrifugation at 10,000 x *g* for 10 min at 4 °C, and denatured in Laemmli buffer before SDS-PAGE. Proteins were transferred to an Immobilon-P PVDF membrane (Millipore), and blocked for two hours in PBST (1 x PBS containing 5 % nonfat dry milk and 0.05 % Tween-20). Membranes were incubated with primary antibodies diluted in PBST overnight at 4 °C and washed extensively in PBST. Membranes were incubated with secondary antibodies for 1 hr at room temperature, washed extensively in PBST, and developed using Immobilon Western Chemiluminescent HRP Substrate (Millipore Sigma). Source data files for all western blots are provided as Source Data File 1.

## STING antibody production

The anti-mbreSTING antibody was generated by Pacific Immunology. Rabbits were immunized with a KLH-conjugated peptide corresponding to residues 320–338 of *M. brevicollis* protein EDQ90889.1 (Cys-KNRSEVLKKMRAEDQYAMP), and serum was affinity purified against the peptide to reduce cross-reactivity and validated using immunoblotting.

## Immunofluorescence staining and imaging

Depending on the cell density of the starting culture, between 0.2 and 1 mL of cells were concentrated by centrifugation for 5 min at 2500× *g*. The cells were resuspended in 200 µl of artificial seawater and applied to poly-ʟ-lysine–coated coverslips (Corning Life Sciences; Cat. No.354085) placed at the bottom of each well of a 24-well cell culture dish. After the cells were allowed to settle on the coverslip for 30 min, 150 µl of the cell solution was gently removed from the side of the dish. All of the subsequent washes and incubations during the staining procedure were performed by adding and removing 200 µl of the indicated buffer.

Cells were fixed in two stages. First, 200 µl cold 6 % acetone diluted in 4 X PBS was added for 5 min at room temperature. Next, 200 µl cold 8 % paraformaldehyde diluted in 4 X PBS was added (yielding a final concentration of 4 % paraformaldehyde), and the fixative mixture was incubated for 15 min at room temperature. After fixation, the coverslip was gently washed three times with 200 µl 4 X PBS.

Cells were permeabilized by incubating in permeabilization buffer (4 X PBS; 3 % [wt/vol] bovine serum albumin (BSA)-fraction V; 0.2 % [vol/vol] Triton X-100) for 30 min. After removing permeabilization buffer, the coverslip was incubated in primary antibody for 1 hour at room temperature, and then washed three times in 4 X PBS. The coverslip was then incubated with secondary antibody for 1 hour at room temperature, and then washed twice in 4 X PBS. The coverslip was next incubated in 4 U/ml Phalloidin (Thermo Fisher Scientific) for 30 min at room temperature, washed once in 4 X PBS. Lastly, the coverslip was incubated in 10 µg/ml Hoechst 33,342 (Thermo Fisher Scientific) for 5 min at room temperature, and then washed once with 4 X PBS.

To prepare a slide for mounting, 10 µl of Pro-Long Gold (Thermo Fisher Scientific) was added to a slide. The coverslip was gently removed from the well with forceps, excess buffer was blotted from the side with a piece of filter paper, and the coverslip was gently placed on the drop of Pro-Long diamond. The mounting media cured overnight before visualization.

Images were acquired on either: (1) a Zeiss LSM 880 Airyscan confocal microscope with a 63 x objective by frame scanning in the superresolution mode (images processed using the automated Airyscan algorithm (Zeiss)), or (2) a Nikon CSU-W1 SoRa spinning disk confocal microscope with a 60 x objective in SR mode (images processed using Imaris).

## Live-cell imaging

Cells transfected with fluorescent reporter plasmid were prepared for microscopy by transferring 200 µl of cells to a glass-bottom dish or glass-bottom 8-well chamber (Ibidi). Confocal microscopy was performed on a Zeiss Axio Observer LSM 880 with an Fast Airyscan detector and a 63 x/NA1.40 Plan-Apochromatic oil immersion objective (Carl Zeiss AG, Oberkochen, Germany). Confocal stacks were acquired by frame scanning in superresolution mode, and images were processed using the automated Airyscan algorithm (Zeiss).

## Transfection of *M. brevicollis*

### Cell culture

One day prior to transfection, 60 ml of High Nutrient Medium was inoculated with *M. brevicollis* to a final concentration of 10,000 cells/ml. The culture was split in two, and grown in two 75 cm² flasks at room temperature, approximately 22 °C (Falcon; Corning, Oneonta, NY, USA; Cat. No. 13-680-65).

### Cell washing

After 24 hr of growth, bacteria were washed away from *M. brevicollis* cells through three consecutive rounds of centrifugation and resuspension in artificial seawater (ASW). The culture flasks were combined and vigorously shaken for 30 s, and then transferred to 50 ml conical tubes and spun for 5 min at 2000× *g* and 22 °C. The supernatant was removed with a serological pipette, and residual media were removed with a fine-tip transfer pipette. The cell pellets were resuspended in a single conical tube in a total volume of 50 ml of ASW, vigorously shaken for 30 s, and then centrifuged for 5 min at 2050× *g*. The supernatant was removed as before. In a final washing step, the cell pellet was resuspended in 50 mL ASW, shaken vigorously, and centrifuged for 5 min at 2100× *g*. After the supernatant was removed, the cells were resuspended in a total volume of 400 µl of ASW. To count the cell density, cells were diluted 100-fold in 200 µl of ASW, and fixed with 1 µl of 16 % paraformaldehyde. Cells were counted on a hemocytometer, and the remaining cells were diluted to a final concentration of $5 \times 10^7$ choanoflagellate cells/ml. The resuspended cells were divided into 100 µl aliquots with $5 \times 10^6$ cells per aliquot to immediately prime cells in the next step.

### Cell priming

Each aliquot of *M. brevicollis* cells was incubated in priming buffer (40 mM HEPES-KOH, pH 7.5; 55 mM lithium citrate; 50 mM L-cysteine; 10 % [wt/vol] PEG 8000; and 2 µM papain) to remove the extracellular material coating the cell. The 100 µl aliquots, which contained $5 \times 10^6$ cells, were centrifuged for 5 min at 1700× *g*. The supernatant was removed, and cells were resuspended in 100 µl of priming buffer and then incubated for 35 min at room temperature. Priming was quenched by adding 4 µl of 50 mg/ml bovine serum albumin-fraction V (Thermo Fisher Scientific, Waltham, MA; Cat. No. BP1600-100) and then centrifuged for 5 min at 1250× *g* and 22 °C with the centrifuge brake set to a "soft" setting. The supernatant was removed with a fine-tip micropipette, and the cells were resuspended in 25 µl of SG Buffer (Lonza).

### Nucleofection

Each transfection reaction was prepared by adding 2 µl of "primed" cells resuspended in SG buffer (Lonza) to a mixture of: 16 µl of SG buffer, 2 µl of 20 µg/µl pUC19, 1 µl of 250 mM ATP (pH 7.5), 1 µl of 100 mg/ml sodium heparin, and ≤7 µl of reporter DNA (volume is dependent on the number of constructs transfected). Each transfection reaction was transferred to one well in 16-well nucleofection strip (Lonza; Cat. No. V4XC-2032). The nucleofection strip was placed in the X-unit (Lonza; Cat. No. AAF-1002F) connected to a Nucleofector 4D core unit (Lonza; Cat. No. AAF-1002B), and the EO100 pulse was applied to each well.

### Recovery

A total of 100 µl of cold recovery buffer (10 mM HEPES-KOH, pH 7.5; 0.9 M sorbitol; 8 % [wt/vol] PEG 8000) was added to the cells immediately after pulsation. After 5 min, the whole volume of the transfection reaction plus the recovery buffer was transferred to 2 ml of Low Nutrient Medium in a 12-well plate. The cells were grown for 24–48 hr before being assayed for luminescence or fluorescence.

### Puromycin selection

To generate stably transfected *M. brevicollis* cell lines, puromycin was added to cells 24 hr after transfection at a final concentration of 300 µg/mL. Cells were monitored over the course of 7–21 days, and fresh High Nutrient Media + 300 µg/mL puromycin was added to the cells as needed.

## Genome editing

For a more detailed description of gRNA and repair oligonucleotide design, refer to *Booth et al., 2018*.

Design and preparation of gRNAs First, crRNAs were designed by using the extended recognition motif 5'-HNNGRSGGH-3' (in which the PAM is underlined, N stands for any base, R stands for purine, S stands for G or C, and H stands for any base except G) to search for targets in cDNA sequences (*Peng et al., 2014*). Next, we confirmed that the RNA sequence did not span exon-exon junctions by aligning the sequence to genomic DNA.

Functional gRNAs were prepared by annealing synthetic crRNA with a synthetic tracrRNA (*Booth and King, 2020*). To prepare a functional gRNA complex from synthetic RNAs, crRNA and tracrRNA (Integrated DNA Technologies [IDT], Coralville, IA, USA) were resuspended to a final concentration of 200 µM in duplex buffer (30 mM HEPES-KOH, pH 7.5; 100 mM potassium acetate; IDT, Cat. No. 11-0103-01). Equal volumes of crRNA and tracrRNA stocks were mixed together, incubated at 95 °C for 5 min in an aluminum block, and then the entire aluminum block was placed at room temp to slowly cool the RNA to 25 °C. The RNA was stored at –20 °C.

Design and preparation of repair oligonucleotides Repair oligonucleotides for generating knock-outs were designed by copying the sequence 50 bases upstream and downstream of the SpCas9 cleavage site. A knockout sequence (5'TTTATTTAATTAAATAAA-3') was inserted at the cleavage site (*Booth and King, 2020*).

Dried oligonucleotides (IDT) were resuspended to a concentration of 250 µM in a buffer of 10 mM HEPES-KOH, pH 7.5, incubated at 55 °C for 1 hr, and mixed well by pipetting up and down. The oligonucleotides were stored at –20 °C.

## Delivery of gene editing cargoes with nucleofection

The method for delivering SpCas9 RNPs and DNA repair templates into *M. brevicollis* is as follows:

### Cell culture

One day prior to transfection, 60 ml of High Nutrient Medium was inoculated to a final concentration of *M. brevicollis* at 10,000 cells/ml. The culture was split in two, and grown in two 75 cm$^2$ flasks at room temperature, approximately 22 °C (Falcon; Corning, Oneonta, NY, USA; Cat. No. 13-680-65).

### Assembly of Cas9/gRNA RNP

Before starting transfections, the SpCas9 RNP was assembled. For one reaction, 2 µl of 20 µM SpCas9 (NEB, Cat. No. M0646M) was placed in the bottom of a 0.25 ml PCR tube, and then 2 µl of 100 µM gRNA was slowly pipetted up and down with SpCas9 to gently mix the solutions. The mixed solution was incubated at room temperature for 1 hr, and then placed on ice.

### Thaw DNA oligonucleotides

Before using oligonucleotides in nucleofections, the oligonucleotides were incubated at 55 °C for 1 hr.

### Cell washing

After 24 hr of growth, bacteria were washed away from *M. brevicollis* cells through three consecutive rounds of centrifugation and resuspension in artificial seawater (ASW). The culture flasks were combined and vigorously shaken for 30 s, and then transferred to 50 ml conical tubes and spun for 5 min at 2000× *g* and 22 °C. The supernatant was removed with a serological pipette, and residual media were removed with a fine-tip transfer pipette. The cell pellets were resuspended in a single conical tube in a total volume of 50 ml of ASW, vigorously shaken for 30 s, and then centrifuged for 5 min at 2050× *g*. The supernatant was removed as before. In a final washing step, the cell pellet was resuspended in 50 mL ASW, shaken vigorously, and centrifuged for 5 min at 2100× *g*. After the supernatant was removed, the cells were resuspended in a total volume of 400 µl of ASW. To count the cell density, cells were diluted 100-fold in 200 µl of ASW, and fixed with 1 µl of 16 % paraformaldehyde. Cells were counted on a hemocytometer, and the remaining cells were diluted to a final concentration of $5 \times 10^7$ choanoflagellate cells/ml. The resuspended cells were divided into 100 µl aliquots with $5 \times 10^6$ cells per aliquot to immediately prime cells in the next step.

## Cell priming

Each aliquot of *M. brevicollis* cells was incubated in priming buffer (40 mM HEPES-KOH, pH 7.5; 50 mM lithium citrate; 50 mM L-cysteine; 15 % [wt/vol] PEG 8000; and 2 µM papain) to remove the extracellular material coating the cell. The 100 µl aliquots, which contained $5 \times 10^6$ cells, were centrifuged for 5 min at 1700× *g* and at room temperature. The supernatant was removed, and cells were resuspended in 100 µl of priming buffer and then incubated for 35 min. Priming was quenched by adding 10 µl of 50 mg/ml bovine serum albumin-fraction V (Thermo Fisher Scientific, Waltham, MA; Cat. No. BP1600-100). Cells were then centrifuged for 5 min at 1250× *g* and 22 °C with the centrifuge brake set to a 'soft' setting. The supernatant was removed with a fine-tip micropipette, and the cells were resuspended in 25 µl of SG Buffer (Lonza).

## Nucleofection

Each nucleofection reaction was prepared by adding 16 µl of cold SG Buffer to 4 µl of the *Sp*Cas9 RNP that was assembled as described above. For reactions that used two different guide RNAs, each gRNA was assembled with *Sp*Cas9 separately and 4 µl of each RNP solution were combined at this step. 2 µl of the repair oligonucleotide template was added to the *Sp*Cas9 RNP diluted in SG buffer. Finally, 2 µl of primed cells were added to the solution with Cas9 RNP and the repair template. The nucleofection reaction was placed in one well of a 16-well nucleofection strip (Lonza; Cat. No. V4XC-2032). The nucleofection strip was placed in the X-unit (Lonza; Cat. No. AAF-1002F) connected to a Nucleofector 4D core unit (Lonza; Cat. No. AAF-1002B), and the EO100 pulse was applied to each well.

## Recovery

100 µl of cold recovery buffer (10 mM HEPES-KOH, pH 7.5; 0.9 M sorbitol; 8 % [wt/vol] PEG 8000) was added to the cells immediately after pulsation. After 5 minutes, the whole volume of the transfection reaction plus the recovery buffer was transferred to 1 ml of High Nutrient Medium in a 12-well plate.

## Cycloheximide selection in *M. brevicollis*

One day after transfection, 10 µl of 10 µg/ml cycloheximide was added per 1 mL culture of transfected cells. The cells were incubated with cycloheximide for 5 days prior to clonal isolation and genotyping.

## Genotyping

Cells were harvested for genotyping by spinning 0.5 ml of cells at 4000 g and 22 °C for 5 min. The supernatant was removed and DNA was isolated either by Base-Tris extraction [in which the cell pellet was resuspended in 20 µL base solution (25 mM NaOH, 2 mM EDTA), boiled at 100 °C for 20 min, cooled at 4 °C for 5 min, and neutralized with 20 µL Tris solution (40 mM Tris-HCl, pH 7.5)], or by DNAzol Direct [in which the cell pellet was resuspended in 50 µL and incubated at room temperature for 30 min (Molecular Research Center, Inc [MRC, Inc.], Cincinnati, OH; Cat. No. DN131)]. Three µl of the DNA solution was added to a 25 µl PCR reaction (DreamTaq Green PCR Master Mix, Thermo Fisher Scientific Cat No K1082) and amplified with 34 rounds of thermal cycling.

**Key resources table**

| Reagent type (species) or resource | Designation | Source or reference | Identifiers | Additional information |
|---|---|---|---|---|
| Strain, strain background (*M. brevicollis*) | *M. brevicollis* | ATCC PRA-258 | PMID:18276888 | |
| Genetic reagent, (*M. brevicollis*) | *M. brevicollis* STING⁻ | This study | | *STING⁻*knockout strain; cell line maintained by A. Woznica |
| Transfected construct (*M. brevicollis*) | pEFL5'-pac-P2A-STING-mTFP-3'act | This study | | Construct to express *M, brevicollis* STING fused to mTFP; can be obtained from A. Woznica |
| Transfected construct (*M. brevicollis*) | pEFL5'-pac-P2A-mCherry-Atg8-–3'act | This study | | Construct to express mCherry fused to *M. brevicollis* Atg8; can be obtained from A. Woznica |
| Strain, strain background (*Flavobacterium*) | *Flavobacterium* sp. | This study | | Isolated from MX1 (ATCC PRA-258) culture; can be obtained from A. Woznica |

*Continued on next page*

*Continued*

| Reagent type (species) or resource | Designation | Source or reference | Identifiers | Additional information |
|---|---|---|---|---|
| Strain, strain background (*Pseudomonas aeruginosa*) | PAO1 | ATCC 15692 | PMID:13961373 | |
| Strain, strain background (*P. aeruginosa*, transgenic strain) | PAO1-GFP | ATCC 15692GFP | PMID:9361441 | |
| Antibody | anti-choano STING (rabbit polyclonal) | This study | | Generated by Pacific Immunology; dilution (1:200) for IF, (1:2000) dilution for WB; can be obtained from A. Woznica |
| Antibody | Anti-mCherry 16D7 (rat monoclonal) | Invitrogen | Cat# M11217 | (1:2000) dilution for WB |
| Antibody | Anti-human Tubulin E7 (Mouse monoclonal) | Developmental Studies Hybridoma Bank | Cat# AB_2315513 | (1:200) dilution for IF |
| Antibody | Alpha-human tubulin (Mouse monoclonal) | Sigma Aldrich | Cat # T64074 | (1:7000) dilution for WB |
| Chemical compound, drug | 2'3' cGAMP | Cayman Chemical | Cat# 19,887 | |
| Chemical compound, drug | 3'3' cGAMP | Cayman chemical | Cat# 17,966 | |
| Sequence-based reagent | *STING556* gRNA | This study | Guide RNA | TTTCGGGATTCAGATGTGGG |
| Sequence-based reagent | *STING* locus PCR primers | This study | PCR primers | F: 5' ATG ATG GTT AAT CTC TCT GAT CTT TCA CAT C 3' R: 5' TTA TGG CAT CGC ATA CTG GTC C 3' |
| Commercial assay or kit | SG Cell Line 4D-NucleofectorTM X Kit S | Lonza, | Cat# V4XC-3032 | |

# Acknowledgements

We thank Bill Jackson, Tera Levin, Shally Margolis, Russell Vance, and Nan Yan for helpful advice and/or comments on the manuscript, and David Greenberg, David Hendrixson, Andrew Koh, Kim Orth, Russell Vance, and Sebastian Winter for bacterial strains. We thank Neal Alto for use of the widefield microscope, Monika Sigg for use of a reporter construct, and Mya Breitbart for sea water samples. We are grateful for David Booth's invaluable advice on developing transfection and gene editing protocols.

This work was funded by a Pew Innovation Fund award (JKP and NK), a HHMI Hanna Gray Fellows award (AW), a HHMI Faculty Scholar award (JKP), and a Burroughs Wellcome Fund Investigators in the Pathogenesis of Infectious Diseases (JKP). We acknowledge the assistance of the UT Southwestern Live Cell Imaging Facility, a Shared Resource of the Harold C Simmons Cancer Center, supported in part by an NCI Cancer Center Support Grant, 1P30 CA142543-01.

# Additional information

## Funding

| Funder | Grant reference number | Author |
|---|---|---|
| Howard Hughes Medical Institute | Hanna Gray Fellows Program | Arielle Woznica |
| Howard Hughes Medical Institute | Faculty Scholars Program | Julie K Pfeiffer |
| Howard Hughes Medical Institute | | Nicole King |

| Funder | Grant reference number | Author |
|---|---|---|
| Pew Charitable Trusts | Pew Innovation Fund | Nicole King<br>Julie K Pfeiffer |
| Burroughs Wellcome Fund | Investigators in the Pathogenesis of Infectious Diseases | Julie K Pfeiffer |
| National Cancer Institute | 1P30 CA142543 | Arielle Woznica<br>Julie K Pfeiffer |

The funders had no role in study design, data collection and interpretation, or the decision to submit the work for publication.

## Author contributions

Arielle Woznica, Conceptualization, Data curation, Formal analysis, Funding acquisition, Investigation, Methodology, Writing - original draft, Writing - review and editing; Ashwani Kumar, Chao Xing, Data curation, Formal analysis; Carolyn R Sturge, Investigation; Nicole King, Funding acquisition, Resources, Writing - review and editing; Julie K Pfeiffer, Conceptualization, Funding acquisition, Supervision, Writing - original draft, Writing - review and editing

## Author ORCIDs

Arielle Woznica ![ORCID] http://orcid.org/0000-0002-3920-1737
Carolyn R Sturge ![ORCID] http://orcid.org/0000-0002-6596-3356
Chao Xing ![ORCID] http://orcid.org/0000-0002-1838-0502
Julie K Pfeiffer ![ORCID] http://orcid.org/0000-0003-2973-4895

## Decision letter and Author response

Decision letter https://doi.org/10.7554/eLife.70436.sa1
Author response https://doi.org/10.7554/eLife.70436.sa2

# Additional files

## Supplementary files

- Transparent reporting form
- Supplementary file 1. Extended key resource tables.
- Source data 1. Extended key resources.

## Data availability

Raw sequencing reads and normalized gene counts have been deposited at the NCBI GEO under accession GSE174340.

The following dataset was generated:

| Author(s) | Year | Dataset title | Dataset URL | Database and Identifier |
|---|---|---|---|---|
| Woznica A | 2021 | STING mediates immune responses in a unicellular choanoflagellate | https://www.ncbi.nlm.nih.gov/geo/query/acc.cgi?acc=GSE174340 | NCBI Gene Expression Omnibus, GSE174340 |

The following previously published datasets were used:

| Author(s) | Year | Dataset title | Dataset URL | Database and Identifier |
|-----------|------|---------------|-------------|-------------------------|
| King N, Westbrook MJ, Young SL, Kuo A, Abedin M, Chapman J, Fairclough S, Hellsten U, Isogai Y, Letunic I, Marr M, Pincus D, Putnam N, Rokas A, Wright KJ, Zuzow R, Dirks W, Good M, Goodstein D, Lemons D, Li W, Lyons JB, Morris A, Nichols S, Richter DJ, Salamov A, Sequencing JG, Bork P, Lim WA, Manning G, Miller WT, McGinnis W, Shapiro H, Tjian R, Grigoriev RD | 2008 | The genome of the choanoflagellate Monosiga brevicollis and the origin of metazoans | https://mycocosm.jgi.doe.gov/Monbr1/Monbr1.home.html | JGI, Monbr1 |

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
