## [Decision Letter]

**Decision letter after peer review:**

Thank you for submitting your article "STING mediates immune responses in the closest living relatives of animals" for consideration by *eLife*. Your article has been reviewed by 3 peer reviewers, and the evaluation has been overseen by a Reviewing Editor and Tadatsugu Taniguchi as the Senior Editor. The following individual involved in review of your submission has agreed to reveal their identity: Ralph Isberg (Reviewer #2).

Essential revisions:

All three reviewers found the work to be of broad interest. The analysis of the response of the choanoflagellate *Monosiga brevicollis* to a variety of bacterial species revealed that exposure of *M. brevicollis* to *Pseudomonas aeruginosa* conditioned medium results in choanoflagellate death and the authors found that his is dependent on the newly discovered ortholog of STING. Characterization reveals that the STING response can be induced by 2'3' cGAMP, which parallels the activation of STING in diverse species. In addition, the finding that cyclic dinucleotide treatment induces autophagy also has parallels with the effector pathways observed in other organisms. There are a number of strengths as outlined by the Reviewers. First, the development of a Choanaoflagellate model system to study innate immunity, second with the development of genetics for M. brevicollis, third, the demonstration of a functional STING system in one of the closest relatives to animals and fourth, that cell death occurs in response to cyclic dinucleotides.

However, there were questions that were raised that if addressed would significantly strengthen the conclusions of the manuscript. One question revolves around cell death, and raises the question of how this cell death pathway is induced and how this controls infection. The second question involves the role of autophagy and whether autophagy is indeed required for STING-dependent antimicrobial defense.

*Reviewer #1 (Recommendations for the authors):*

A) The authors should address why only *Pseudomonas aeruginosa* is causing the generation of 2'3' cGAMP. Are these CDNs found in the conditioned media (Figure 1H-I), or do the authors hypothesize that an unidentified *Pseudomonas* factor is responsible for the generation of 2'3' cGAMP when the media is exposed to *M. brevicollis*? On the other hand, if they hypothesize that the *M. brevicollis* cGAS ortholog is responsible for the generation of 2'3' cGAMP, they could test this hypothesis using a cGAS null mutant.

B) Figure 6 – Here, the authors should include data showing that Atg8 lipidation and the induction of autophagy leads to M*. brevicollis* death. They could use their chloroquine treatment method to test if this inhibits *Pseudomonas*-induced, STING-mediated killing of *M. brevicollis*.

*Reviewer #2 (Recommendations for the authors):*

1. I would recommend the authors drop the figures on phagocytosis resistance to *Pseudomonas*. The *Pseudomonas* type 3 secretion system has a number of effectors that antagonize phagocytosis, so it's unclear if this is a host or bacterial activity. In any event, unless it is directly linked to STING, it's probably not relevant to the manuscript.

2. I was a little confused about the cell supernatant strategy. Are the authors saying that contact with bacteria and phagocytosis are unable to induce the response, and you need a secreted molecule? It might have helped if they did some quick and dirty experiments, like heating the supernatant, or size fractionating to argue for a small molecule.

3. This may be major or minor for 1F: the MOIs are unclear and this wasn't done in a fashion in which it was easy to understand what was going on (see above critique). To be done carefully, different MOIs have to be used and determination if TTSS is involved, etc. I really think this doesn't contribute to the story and should be dropped.

4. Figure 4: Given that you raised antibody, why did you use transfected cells here to analyze STING localization?

5. Line 232 is a heading, not a sentence.

6. I am not convinced M. fluctuans is devoid of STING. It has a cGAS and interestingly has an NFkB. Have you tried seeing if it responds to 2'3'cGAMP?

7. The authors don't give a model for the death response. Are they proposing that there is an inflammasome-like function that STING is controlling? The fact that LPS also seems to induce a parallel pathway suggests that there is a pattern recognition-generated cellular suicide (which would primarily make sense for intracellular pathogens). They also might provide some rationale for why cell death protects against an extracellular pathogen (unless there is an intracellular lifestyle for *Pseudomonas*).

*Reviewer #3 (Recommendations for the authors):*

1. Quantitation of flagella length (Figure 1B) is missing

2. Are any *P. aeuginosa* effects altered in the STING KO strain?

3. I think that more experimental inquiry is needed to determine the potentially distinct *P. aeruginosa*-induced signaling pathway and the environmentally relevant trigger 0f the cGAS/STING response.

4. I would like a stronger evolutionary argument as to the significance of the system. It appears to me that Choanaflagellates were data-mined to find innate immune homologs such as STING. While two (of >20) species have conserved cGas/STING pairs, is there evidence that this is evolutionarily relevant to animals?

---

## [Author Response]

Essential revisions:All three reviewers found the work to be of broad interest. The analysis of the response of the choanoflagellate Monosiga brevicollis to a variety of bacterial species revealed that exposure of M. brevicollis to *Pseudomonas aeruginosa* conditioned medium results in choanoflagellate death and the authors found that his is dependent on the newly discovered ortholog of STING. Characterization reveals that the STING response can be induced by 2'3' cGAMP, which parallels the activation of STING in diverse species. In addition, the finding that cyclic dinucleotide treatment induces autophagy also has parallels with the effector pathways observed in other organisms. There are a number of strengths as outlined by the Reviewers. First, the development of a Choanaoflagellate model system to study innate immunity, second with the development of genetics for M. brevicollis, third, the demonstration of a functional STING system in one of the closest relatives to animals and fourth, that cell death occurs in response to cyclic dinucleotides.However, there were questions that were raised that if addressed would significantly strengthen the conclusions of the manuscript. One question revolves around cell death, and raises the question of how this cell death pathway is induced and how this controls infection.

We performed additional experiments and found that cell death requires sustained exposure to *P. aeruginosa*. Based on transcriptional profiling, we hypothesize that cell death may be the result of overstimulating stress response pathways. These new results can be found in Figure 1J and are described in lines 203-220. These observations will be the foundation of an area of future investigation.

The second question involves the role of autophagy and whether autophagy is indeed required for STING-dependent antimicrobial defense.

Because *P. aeruginosa* has multiple factors that induce different stress response pathways in *M. brevicollis*, it is difficult to separate STING-specific effects on autophagy from other pathways that induce autophagy (and other host responses). We have done further experiments investigating autophagy with 2’3’ cGAMP since it allows us to examine STING-specific effects. We now show that autophagy is required for 2’3’ cGAMP-induced cell death. These new results are found in Figure 6H, and summarized in lines 414-422. Autophagy in the context of whole/complex bacterial cells will be an area of future investigation.

Reviewer #1 (Recommendations for the authors):A) The authors should address why only *Pseudomonas aeruginosa* is causing the generation of 2'3' cGAMP. Are these CDNs found in the conditioned media (Figure 1H-I), or do the authors hypothesize that an unidentified Pseudomonas factor is responsible for the generation of 2'3' cGAMP when the media is exposed to M. brevicollis? On the other hand, if they hypothesize that the M. brevicollis cGAS ortholog is responsible for the generation of 2'3' cGAMP, they could test this hypothesis using a cGAS null mutant.

Thank you for raising these points. Because secreted virulence factor(s) produced by *P. aeruginosa* induce a broad transcriptional host response in *M. brevicollis,* it difficult to narrow down why *P. aeruginosa*, but not other bacteria*,* induces STING upregulation. We do not think that CDNs produced by *P. aeruginosa* are responsible for the cytotoxic effects induced by *P. aeruginosa,* or for the upregulation of STING, because known bacterially-produced CDNs (3’3’ cGAMP, c-di-GMP, c-di-AMP) do not induce cell death or STING upregulation in *M. brevicollis* (Figure 3C, 3D). Although it is beyond the scope of this study, we hope that future characterization of the chemical structures of these *P. aeruginosa* virulence factors will help us to better understand why *P. aeruginosa*, but not other bacteria (for instance, those also harboring active T3SS), is pathogenic towards *M. brevicollis*.

We have clarified in the Discussion (lines 445-454) the possibility that *M. brevicollis* cGAS may be responsible for the production of an endogenous CDN, like 2’3’ cGAMP that activates STING.

“For example, while our results demonstrate that *M. brevicollis* STING responds to exogenous 2’3’ cGAMP, the endogenous triggers of STING activation in *M. brevicollis* remain to be determined. […] Although cGAS and STING are rare among sequenced choanoflagellate species, both species with STING homologs, *M. brevicollis* and *S. macrocollata*, also harbor a cGAS homolog (Figure 1- supplementary figure 1), suggesting the presence of an intact choanoflagellate cGAS-STING pathway.”

We agree that extensive in vitro biochemical characterization of *M. brevicollis* cGAS, and phenotypic characterization of a cGAS null mutant will help substantiate or refute this hypothesis. While *M. brevicollis* reverse genetics are now possible due to our efforts for this paper, the process remains slow and labor intensive. Thus, while characterizing a cGAS null mutant was not feasible for this paper, it will be a focus of future work.

B) Figure 6 – Here, the authors should include data showing that Atg8 lipidation and the induction of autophagy leads to M. brevicollis death. They could use their chloroquine treatment method to test if this inhibits Pseudomonas-induced, STING-mediated killing of M. brevicollis.

Thank you for this wonderful suggestion. We have added Figure 6H showing that pretreatment with lysosomotropic agents chloroquine or NH_4_Cl rescues cell death induced by 2’3’ cGAMP. This data has led to the hypothesis that a primary mechanism by which 2’3’ cGAMP induces STING-mediated cell death is via autophagic signaling, and is summarized in lines 414-422.

“Finally, we asked whether STING-mediated autophagic pathway induction affects survival after exposure to 2’3’ cGAMP (Figure 6H). […] Therefore, we hypothesize that 2’3’ cGAMP induces cell death in *M. brevicollis* by overstimulating STING-mediated autophagic signaling.”

In addition, we agree that a more complete understanding of how *P. aeruginosa* induces cell death would strengthen this manuscript, and have modified lines 504-507 the Discussion to iterate this:

“Thus, identifying specific *P. aeruginosa* virulence factors will be critical for understanding why *P. aeruginosa* – but not other bacteria – have pathogenic effects on *M. brevicollis*, and facilitate the characterization of mechanisms underlying choanoflagellate pathogen responses.”

Finally, because *P. aeruginosa* leads to broad host responses, including the induction of many stress-related genes, we have not explored STING-mediated autophagy in response to *P. aeruginosa* in this manuscript. We hope that building additional tools in *M. brevicollis* to study autophagy, along with identifying specific *P. aeruginosa* virulence factors, will enable us to address the role of autophagy in choanoflagellate antibacterial immunity in the future.

Reviewer #2 (Recommendations for the authors):1. I would recommend the authors drop the figures on phagocytosis resistance to Pseudomonas. The Pseudomonas type 3 secretion system has a number of effectors that antagonize phagocytosis, so it's unclear if this is a host or bacterial activity. In any event, unless it is directly linked to STING, it's probably not relevant to the manuscript.

Thank you for this recommendation. We agree that we didn’t include sufficient data to support whether our observation that *M. brevicollis* does not internalize *P. aeruginosa* is due to host or bacterial factors. Nonetheless, we find this observation to be important because it suggests that internalization of *P. aeruginosa* is not required for it to exert pathogenic effects.

We have edited the text to emphasize that phagocytosis of *P. aeruginosa* is not required to exert pathogenic effects, and have removed wording suggesting that *M. brevicollis actively* avoids phagocytosis of *P. aeruginosa*. These edits can be found in lines 158-181.

We have also performed additional experiments to examine *P. aeruginosa* T3SS mutants induce cell death in *P. aeruginosa.* Experiments with both live bacteria and conditioned media found that deletion strains ∆popB and ∆exoSTY induce death similarly to wild type (strains listed in Table 2). While we do not explore these results in the text, we thought they might be of interest to you.

2. I was a little confused about the cell supernatant strategy. Are the authors saying that contact with bacteria and phagocytosis are unable to induce the response, and you need a secreted molecule? It might have helped if they did some quick and dirty experiments, like heating the supernatant, or size fractionating to argue for a small molecule.

Thank you for this comment. We agree that we could have more clearly stated our rationale for testing cell supernatant. We have added additional text describing our rational for testing cell supernatant in lines 180-183.

“The above results suggested that the pathogenic effects of *P. aeruginosa* are induced by factors secreted by extracellular bacteria. In addition, diverse secreted bacterial molecules have been previously shown to influence choanoflagellate cell biology.”

We also added additional text in lines 195-202 and a simple table (Table 3) supporting the hypothesis that secreted small molecules induce cell death in *M. brevicollis:*

“The bioactivity in the conditioned media was also found to be heat, protease, and nuclease resistant, indicating that the virulence factors are unlikely to be proteins or nucleic acids (Table 3). […] Although further detailed chemical analysis is required to determine the molecular nature of these factors, these data indicate that secreted *P. aeruginosa* small molecules are sufficient for inducing cell death in *M. brevicollis.”*

3. This may be major or minor for 1F: the MOIs are unclear and this wasn't done in a fashion in which it was easy to understand what was going on (see above critique). To be done carefully, different MOIs have to be used and determination if TTSS is involved, etc. I really think this doesn't contribute to the story and should be dropped.

Thank you for this suggestion. We have added details regarding MOI into the Figure legend to make this experiment more clear, and additional methods regarding exact choanoflagellate and bacterial cell densities can be found in the Methods. In addition, we have tried to make clear in lines 161-171 that our conclusion is only that *M. brevicollis* does not internalize *P. aeruginosa* but does internalize other bacteria tested.

4. Figure 4: Given that you raised antibody, why did you use transfected cells here to analyze STING localization?

In Figure 3- supplementary figure 1E,F and lines 280-281, we show microscopy of endogenous STING staining in untreated cells and cells stimulated with 2’3’ cGAMP. We found that the pattern of STING staining using the antibody looked similar to that of tagged STING in transfected cells. However, without antibodies or live stains that reliably highlight organelles in *M. brevicollis,* we could not further address the localization of STING using IF. For this reason, we chose to co-transfect STING with markers that illuminate organelles. We are currently working on methods to tag endogenous genes in *M. brevicollis*, which may provide better resolution in the future.

“In addition, immunostaining for STING in fixed *M. brevicollis* revealed that the number and intensity of STING puncta increases after exposure to 2’3’ cGAMP (Figure 3- supplementary figure 1E,F), although the localization of STING was difficult to assess by immunostaining due to a lack of available subcellular markers”.

5. Line 232 is a heading, not a sentence.

We agree that this statement was unnecessary, and have deleted it.

6. I am not convinced M. fluctuans is devoid of STING. It has a cGAS and interestingly has an NFkB. Have you tried seeing if it responds to 2'3'cGAMP?

We agree that it is interesting that *M. fluctuans* has putative cGAS and NFkB proteins, but not STING. The transcriptome of *M. fluctuans* was sequenced by Richter et al., 2018, and evidence for gene presence was evaluated using BLAST-based approaches. We could not find evidence for a TMEM173 domain (indicative of a STING homolog) in *M. fluctuans*. Unfortunately, we have had difficulties recovering *M. fluctuans* from cryopreserved stocks*,* and haven’t been able to test whether it responds to 2’3’ cGAMP. We would really like to do this experiment!

7. The authors don't give a model for the death response. Are they proposing that there is an inflammasome-like function that STING is controlling? The fact that LPS also seems to induce a parallel pathway suggests that there is a pattern recognition-generated cellular suicide (which would primarily make sense for intracellular pathogens). They also might provide some rationale for why cell death protects against an extracellular pathogen (unless there is an intracellular lifestyle for Pseudomonas).

We agree that it is interesting that *M. fluctuans* has putative cGAS and NFkB proteins, but not STING. The transcriptome of *M. fluctuans* was sequenced by Richter et al., 2018, and evidence for gene presence was evaluated using BLAST-based approaches. We could not find evidence for a TMEM173 domain (indicative of a STING homolog) in *M. fluctuans*. Unfortunately, we have had difficulties recovering *M. fluctuans* from cryopreserved stocks*,* and haven’t been able to test whether it responds to 2’3’ cGAMP. We would really like to do this experiment!

Reviewer #3 (Recommendations for the authors):1. Quantitation of flagella length (Figure 1B) is missing

Thank you for this recommendation, we have now included quantification in Figure 1B.

2. Are any P. aeuginosa effects altered in the STING KO strain?

In STING KO cells, exposure to *P. aeruginosa* initially restricts growth (Figure 5I,J). However, in contrast to WT cells in which cell growth is completely restricted, STING KO cells are still able to divide in the presence of *P. aeruginosa*, and largely recover from early growth inhibition:

“We also examined the survival of *STING­^–^* cells exposed to *P. aeruginosa* conditioned medium (Figure 5I,J). […] These results indicate that wild type cells are more susceptible to *P. aeruginosa* than *STING^–^* cells, although it is unclear how STING contributes to *P. aeruginosa*-induced growth restriction and cell death.”

3. I think that more experimental inquiry is needed to determine the potentially distinct *P. aeruginosa*-induced signaling pathway and the environmentally relevant trigger 0f the cGAS/STING response.

Thank you for this suggestion. We concur that much more work is required to determine (i) the signal transduction pathways by which *M. brevicollis* responds to *P. aeruginosa,* and (ii) the environmental and endogenous molecules that activate cGAS/STING response. While we fully intend to explore these research questions in the future, such questions will likely take years to tackle in our choanoflagellate system and are beyond the scope of the current study. In the discussion (lines 404-415, and lines 457-468) , we try to address many these questions that will be imperative to explore to understand the mechanisms by which STING contributes to antibacterial immunity in *M. brevicollis*.

4. I would like a stronger evolutionary argument as to the significance of the system. It appears to me that Choanaflagellates were data-mined to find innate immune homologs such as STING. While two (of >20) species have conserved cGas/STING pairs, is there evidence that this is evolutionarily relevant to animals?

Thank you for this comment. Choanoflagellates have served as key models for reconstructing the genetic and developmental foundations of animal origins. As the closest living relatives of animals, comparing features shared by extant lineages of choanoflagellates and animals can provide unique insight into features that were likely present in the last common ancestor of animals, including features of innate immunity.

While choanoflagellates were not “data-mined” to find innate immune homologs, the most powerful approach we have for inferring genes that were present in the last common ancestor of animals is comparative genomics. Through comparative genomics, Richter et al. (2018) were able to identify nearly 400 gene families – including some innate immune genes, like cGAS and STING – that evolved prior to the divergence of animals and choanoflagellates.

We did not initially set out to study STING; however, when we found that STING was upregulated in *M. brevicollis* in response to *P. aeruginosa* (lines 193-196)*,* we were excited to further explore its evolutionary history. Choanoflagellate and animal STING proteins are similar both structurally and at the amino acid level, and STING homologs have not been detected in any other lineages closely related to choanoflagellates or animals. Thus, we believe that STING found in animals today arose in a clade containing choanoflagellates and animals (lines 73-74), and that studying STING in choanoflagellates can provide insight into the evolutionary history of this protein in animals.